# Effect of Eccentricity of Applied Force and Geometrical Imperfections on Stiffness of Stack of Cuboidal Steel Elements

**DOI:** 10.3390/ma13143174

**Published:** 2020-07-16

**Authors:** Mateusz Smolana, Krzysztof Gromysz

**Affiliations:** 1Department of Mechanics and Bridges, Faculty of Civil Engineering, Silesian University of Technology, Akademicka 5, 44-100 Gliwice, Poland; mateusz.smolana@polsl.pl; 2Department of Building Structures, Faculty of Civil Engineering, Silesian University of Technology, Akademicka 5, 44-100 Gliwice, Poland

**Keywords:** building rectification, eccentricity, initial geometric imperfection, internal friction, stiffness, support, hysteresis loop

## Abstract

Experimental tests were performed on a stack of cuboidal steel elements and its mathematical model was developed. Each cuboidal element was made of rolled profiles. Such stacks are the part of temporary supports that can be used to eliminate deflections of buildings. Stacks were loaded eccentrically through the inaccurate position of a jack. Moreover, two types of geometrical imperfections could be noticed. They included inaccurate contact between the stack elements and initial relative displacements of profiles that formed cuboidal elements. A mathematical model was developed to describe deformations of the stack and its parameters were determined by analysing test results. The eccentricity of the applied force had a slight impact on the stack stiffness, which was considerably reduced by geometrical imperfections. The imperfection covering initial relative displacements of rolled profiles inside cuboidal elements had the greatest impact on the stiffness. It could cause even a 10-fold drop in the stack stiffness when compared with the theoretical stiffness resulting from the stiffness of the stack section, and the stiffness dropped by ca. 3.5 times when the imperfection included the inaccurate contact between the cuboidal elements. Finally, the occurrence of both types of geometrical imperfections generated the real stiffness more than ten times lower than the theoretical stiffness that did not take into account imperfections.

## 1. Introduction

Stacks of cuboidal elements are commonly used to rectify vertically deflected buildings. Rectification consists of installing hydraulic piston jacks in walls of a building cellar [1]. Jacks cause the detachment of a building. The detached surface runs through all elements of a building and divides it into two parts: an unevenly lifted part and a part remaining in the ground (Figure 1a). Such uneven lifting, which is to put the lifted part into vertical position, consists in proper extending of jack pistons. As the extension of the jack pistons is limited to 200 mm, a stack of cuboidal elements made of steel have to be installed under them (Figure 1b). Therefore, temporary supports of the building are created. They consist of a jack and a stack of cuboidal elements made of steel. If the jack is inaccurately put on the stack, then the jack axis does not overlap with the axis of the stack of cuboidal elements. Thus, the unintentional eccentricity *e*_a_ is observed (Figure 2a).

Cuboidal elements being components of tested stacks, are composed of five rolled profiles: three profiles UPN160 and two halves of profiles UPN160 (Figure 2b). They are connected with edge welds that are not structural welds. This type of weld is dedicated to stabilising the relative position of rolled profiles. The dimensions of one element are 350 × 320 × 72.5 mm (Figure 2b).

Supporting surfaces of cuboidal elements are not ideal planes. Hence, the stack elements do not always strictly adhere to them (Figure 2c). Moreover, relative displacements were observed for rolled profiles forming cuboidal elements. Observable defects were called geometrical imperfections. Two types of imperfections were distinguished. The first type included the inaccurate contact between the cuboidal elements, and the second one referred to initial relative displacements of profiles inside the cuboidal elements. They were the consequences resulting from deviations in dimensions of rolled profiles and assumed constructional solutions for cuboidal elements.

During rectification, the weight of the unevenly lifted part of the building rests on temporary supports. Thus, those supports, particularly their stiffness in the longitudinal direction, determine the safety of the process. Therefore, tests were developed and conducted to determine the stiffness of the stack depending on values of unintended eccentricity. The aim of the tests was also to determine the effect of two observable geometrical imperfection on the stack stiffness.

The stack of unconnected cuboidal elements transmits only compressive stress, and tensile stresses cannot be compensated. Similar systems had been already tested with respect to lost stability of a stack composed of rigid bodies. *Block-stacking problem* is a classic issue that emerged in the 19th century [2] and concerned the maintenance of the static balance of the stack, in which each successive element was displaced with respect to the preceding one. It was demonstrated the stack made of elements displaced with respect to each other could be higher if friction forces were acting between the elements [3]. That problem was developed by, among others, the construction of stacks, where one layer was composed of at least two elements or the interaction of more than one stack [4]. With reference to building structures, tests were conducted on, among others, precast reinforced concrete columns composed of the stack of elements. The bearing capacity of a stack of defined height was demonstrated to decrease along with an increasing number of elements forming that stack [5]. The test described in the paper [6] found that reduced transverse strain in elements of the stack increased its bearing capacity. It was demonstrated in the paper [7] that supporting surfaces of the stack element should be inclined at the angle narrower than 30 degrees. Issues related to stacks are also observed in masonry structures with dry joints. For example, calculations were performed on using homogenisation techniques for the masonry wall [8], or dry joints in refractory linings were found to reduce stresses induced by temperature by 50% [9]. Other tests demonstrated an increase in the temperature of the stack used as refractory linings increased their stiffness [10]. A drop in the modulus of elasticity, of the material used for joints resulted in reduced bearing capacity of the stack of elements [11]. Non-destructive methods for determining the mechanical properties of material used for joints in masonry elements are presented in [12]. The paper [13] proposed a numerical model of refractory mortarless masonry. Tests on that structure using Digital Image Correlation demonstrated the joint width between elements subjected to loading decreased [14]. The available literature also contains numerous papers presenting tests on mortarless masonry walls made of small elements. Experimental tests showed the bearing capacity of that structure depend on the slenderness of the wall [15] or column [16], and on the value of eccentricity of applied load [17,18,19]. Analytical calculations [20] presented the effect of calculated bearing capacity of the element stack on the assumed non-linear model of the material used in elements forming the stack.

Stability of no-tension elements under compression was tested in the paper [21] by introducing an equivalent section of a rod corresponding to the compressed part of the structure. The critical loading for such systems was determined in the paper [22]. Theoretical analyses and experimental tests on stacks made of stone elements were described in the paper [23]. This paper confirmed analytical relationships for a critical load, based on experimental tests.

There are few tests on supports composed of the jack and a stack of cuboidal elements. However, they only referred to axially compressed supports of stacks made of cuboidal wooden [24] and steel [25] elements. Moreover, such supports examined in situ [26] showed their stiffness has a decisive impact on parameters of a rectified building.

Stacks of cuboidal elements were used, among others, to eliminate deflection of an 11-storey residential building [27], a five-storey residential building [26], a historic tower [28], and a single-family house [29]. Moreover, the plan was developed to eliminate deflection of a historic neo-Gothic church of a weight of 900 tons [30].

Deflections of buildings are commonly observed. It is caused by the appearance of the effect of underground coal mining [31,32], exploitation of natural groundwater [33], compaction of loess [34] and earthquakes [35]. The theory of rock movement is employed to forecast deflections of structures caused by mining [36,37].

## 2. Research Programme

A stack of the length *l* equal to 1.0 m made of cuboidal steel elements was the subject of the research (Figure 3a). The stack was a component of a temporary support, in which the axis of the hydraulic jack did not overlap with the stack axis. The unintentional eccentricity *e*_a_ between those elements was in the plane *z*-*x*. One purpose of the research was to determine the stiffness of the stack installed in the temporary support and test the impact of the eccentricity value *e*_a_ on that stiffness. Another purpose was to analyse the impact of geometrical imperfections of cuboidal elements on the stack stiffness. The imperfections included inaccurate contact between adjacent cuboidal elements and initial relative displacements of rolled profiles forming cuboidal elements.

In real conditions, the temporary support rests on steel plate installed in concrete grouting, which prevents the support base from displacement or rotation. A piston of the hydraulic jack is ended with a hinge so the bending moment is not transferred on the support. In a particular case, a one-end-fixed rod loaded with concentrated force could be considered as the static scheme of the support [38] (Figure 3b).

That support scheme under the gravitational load *Q* equal to 1000 kN could not be reproduced under laboratory conditions due to safety standards, therefore, an equivalent system was used. It comprised two identical supports, the bases of which were directed to each other (Figure 3c). The analysed system consisted of a stack of twenty-seven cuboidal elements made of steel with a length 2*l* equal to 2.0 m, and two jacks. A free-ends rod of double the length of one-end fixed rod was used as a diagram of the tested system. Deformations of the analysed system represented deformation of the support loaded with the concentrated force *Q* at eccentricity *e*_a_ (Figure 3d). The tested system was installed in a horizontal position at the test stand that consisted of a steel frame, a platform made of polished steel plate, and ball transfer units (Figure 4a). The system was rested on ball transfer units placed directly on the platform plate (Figure 4b). They were used to minimise friction between the stack and the platform.

The force *Q* was induced by increasing the pressure of oil in the active jack. Its value was measured using the compression force transducer installed between the passive hydraulic jack and the steel frame. Moreover, changes in length Δ*l*_A_ and Δ*l*_B_ of both edges of the stack were measured (Figure 5). A change in the length Δ*l* of the stack along the axis of the applied load, taking into account the presence of eccentricity *e*_a_, was determined by linear interpolation between the values measured at the edges of the stack:(1)$$\Delta l=\frac{2{\Delta l}_{A}+2\Delta l_{B}}{4}+\frac{2\Delta l_{A}-2\Delta l_{B}}{4\times 320}e_{a}.$$

The measuring base used to determine changes in length Δ*l*_A_ and Δ*l*_B_ was attached to a stack of rectangular elements and had no contact with frame. So, the frame rigidity did not affect the measured values of Δ*l*_A_ and Δ*l*_B_.

Cyclic tests were designed for the system loaded with the force *Q*, in which the maximum load (*Q*_max_) and the unintentional eccentricity (*e*_a_) values were variable parameters. Each cycle consisted of loading and unloading the system four times with *Q* values from 0 to *Q*_max_ (Figure 6). Depending on the measurement, *Q*_max_ of one of the following values was forced: 100, 250, 500, 750 and 1000 kN. Values of the unintentional eccentricity *e*_a_ were as follows: 0, 10, 20, 30 and 40 mm. Each of the test cycles was named to identify its parameters (Table 1). All tests were originally intended to be performed under the load of 1000 kN. However, it was found during preliminary tests that the maximum force of 750 kN could be applied at the eccentricity *e*_a_ equal to 20 mm, for the eccentricity of 30 mm the maximum applied force was 500 kN, and for the eccentricity equal to 40 mm, the maximum applied force was only 100 kN. Attempts to apply higher forces failed as the stack stability was lost.

## 3. Test Procedure and Results

Each test was preceded with a precise, and coaxial arrangement of the stack elements and the arrangement of hydraulic jacks in conformity with the assumed eccentricity *e*_a_ for a given test. The test included measuring changes in the length 2Δ*l*_A_ and 2Δ*l*_B_ of the stack edge under increasing and decreasing force *Q* and recording them at a frequency of 2 Hz. Four full load–unload cycles in each test were performed during which the longitudinal force was monotonically changing its value from close to zero (1.5 kN) to *Q*_max_, and then was decreasing to the value close to zero. The force values *Q* within the range from 0 to 1.5 kN were neglected during the tests because the elements have not been adjacent to each other under that load yet. The reason was their loose arrangement. The test stand is shown in Figure 7a–c to illustrate ball transfer units and a platform on which the tested system was placed. It is worth mentioning here that in a real situation, at the construction side, the stack of elements is installed in a vertical position. The dead weight of a 1 m high stack is approximately 3 kN. The values of force *Q* within the range of 0 to 1.5 kN were neglected during the test, taking under consideration the dead weight of the stack.

Selected test results are shown in Figure 8 illustrating recorded cycles *Q*-Δ*l* during which the lowest and the highest *Q*_max_ value corresponding to the given eccentricity *e*_a_ was observed. Each diagram presents four load–unload cycles. The system returned each time to its initial position, and hysteresis loops were formed. Loops corresponding to a given load *Q*_max_ and eccentricity *e*_a_ overlapped with each other. For this reason, the single hysteresis loop from each test was considered as representative for further analysis. Thus, diagrams in Figure 9 illustrate the summary of *Q*-Δ*l* relationships for a given eccentricity *e*_a_ limited to singular hysteresis loops.

The graphs presented in Figure 9 had common characteristics. Three phases were identified for each *Q*-Δ*l* cycle. Phase *I* corresponded to the monotonous increase in the load from zero to *Q*_max_ (Figure 9f). The relationship *Q*-Δ*l* in that phase was non-linear as a clear increase in the slope of the curve and load was observed. Unloading at first caused a rapid drop in the force value at the slightly changed length of the stack. In that case, the relationship *Q*-Δ*l* was nearly linear. That part of a diagram was marked as Phase *II*. Then, the further drop in the load was accompanied by a less intensive decline in the force value *Q* and change in the length Δ*l*. At the same time, the relationship *Q*-Δ*l* became definitely non-linear. Furthermore, the relationships *Q*-Δ*l* overlapped with each other in Phases *I* and *III* for a given value of the eccentricity regardless of the maximum load value. For various values of *Q*_max_ in the system *Q*-Δ*l,* the line slope corresponding to Phase *II* changed.

## 4. Model of the Stack

The aim of building the mathematical model of the stack was to determine the relationship between load and displacement (*Q*-*u*) of the system composed of cuboidal elements. The defined model of the stack should represent load-displacement (*Q*-Δ*l*) relationships observed during tests conducted on the real stack. A structure of the model was defined at first, and then the equations describing its particular elements were used.

### 4.1. Structure of the Model

The model was defined on the basis of observations made during analyses of the tests. The real stack under loading equal to the maximum value *Q*_max_ and then unloading was found to return to its initial position. Thus, there were only elastic deformations of the material used in the stack. However, parts of the relations *Q*-Δ*l* were not straight lines, which meant the stack stiffness was not constant. Hence, modelling of the stack element could not only be limited to the use of linear-elastic elements. Moreover, the occurrence of the hysteresis loop indicated the presence of non-conservative friction forces in the real system.

Displacement of the loaded end of the stack was expressed as *u*_st_ and was caused by elastic deformations of the material occurring in the direction of the load (Figure 10a). That displacement showed a linear dependence on the load *Q*, and in the model, it corresponded with deformations of a spring with stiffness *k*_st_. The stiffness was identified with the rod under axial compression. The stiffness *k*_st_ was thus constant and included deformations of the first order. It did not include a structural solution for the stack containing cuboidal elements made of five rolled profiles each (Figure 2b). The effect of the structural solution on the system stiffness could be evaluated by referring the real stiffness of the stack to the stiffness *k*_st_.

Analysis of individual cuboidal elements of the real stack indicated the presence of geometrical imperfections (cf. Figure 2c). The imperfections could be divided into two groups. The first group included inaccurate contact between cuboidal elements from the stack. Their surfaces exposed to loading are closing to each other as the result of bending and twisting of the cuboidal elements. Those displacements are marked as *u*_con_ (Figure 10b). The interface of adjacent elements changes due to increasing loading. It means that displacement values *u*_con_ were not linearly dependent on the load value *Q*. The non-linear stiffness *k*_con_ corresponds to displacements *u*_con_ in the model.

The second group of imperfections included initial relative displacements of rolled profiles forming cuboidal elements. Those imperfections were caused by both deviations in dimensions of sections of profiles UPN160, and production inaccuracies of cuboidal elements. Deformations of cuboidal elements occurred as the load was increasing. They were non-dilatational strains of rolled profiles forming cuboidal elements and deformations of welds joining rolled profiles. Consequently, relative displacements of rolled profiles forming cuboidal elements are observed and denoted by *u*_int_ (Figure 10c). Those displacements were in the direction of the load *Q*. An increased load was likely to cause displacements *u*_int_ of some cuboidal elements, but not all of them. The displacement was also non-linearly depending on the load. Hence, the spring *k*_int_ representing that displacement did not have the constant stiffness. Deformations of profiles corresponding to the displacements *u*_int_ resulted in the mutual contact pressure of profiles forming cuboidal elements. The pressure was in the direction perpendicular to the load *Q*. Internal friction forces, denoted by *N*_int,fr_, were generated at the interface between profiles displacing with respect to each other. Forces acted in the direction of the load *Q* and were responsible for the formation of the hysteresis loop observed during the tests.

Displacement of the model was the sum of displacements of components. Thus, it could be expressed as follows:(2)$$u=u_{\mathrm{st}}+u_{\mathrm{cont}}+u_{\mathrm{int}}$$

Each of the displacement components was the sum of displacements related to each *i*-th cuboidal element. As the stack was composed of *n* elements, it could be expressed as follows (Figure 10):(3)$$u_{\mathrm{st}}=\overset{n}{\underset{i=1}{\sum }}u_{\mathrm{st},i},\;\;u_{\mathrm{con}}=\overset{n}{\underset{i=1}{\sum }}u_{\mathrm{con},i},\;u_{\mathrm{int}}=\overset{n}{\underset{i=1}{\sum }}u_{\mathrm{int},i}.$$

The displacement *u* of the model expressed as (2) corresponded to the change Δ*l* in the length of the real stack. As previously described, each of the displacement components is related to deformations of the material used in the stack. Those deformations are considerably smaller than the plastic deformations. Components *u*_st_, *u*_con_, *u*_int_ of the displacement in the model were elastic displacements. Each of those displacements corresponded to spring deformations in the model that are denoted by *k*_st_, *k*_cont_ and *k*_int_. Due to (2), the springs were connected in series, as illustrated in Figure 11a. Moreover, there were friction forces *N*_int,fr_ in the model, which represented internal friction.

The quasi-static load exerted on the stack did not generate internal friction in the material used to create the stack. There was no likelihood of friction on supporting surfaces in the direction of the load *Q*. Friction revealed if the displacement *u*_int_ occurred. Therefore, the spring *k*_int_ and the element, in which friction *N*_int,fr_ occurred, were connected in parallel. A pair of elements *k*_int_ and *N*_int,fr_ represented the behaviour of the cuboidal element, and the spring *k*_con_ specified deformability of the interface between the cuboidal elements. They are elements of the model representing the structural solution for the stack. The spring *k*_st_ is responsible for stiffness of the stack, for which the structural solutions are not included. Mathematical relationships describing particular spring and internal frictions are described in subsections below.

### 4.2. Effect of the Number of Cuboidal Elements on the Stack Stiffness

The stiffness *k*_st_ corresponded to the stiffness of the stack in the form of a homogeneous rod under compression. Its value was constant and determined as the ratio of change Δ*Q* of the load value to the adequate change Δ*u*_st_ in the displacement:(4)$$k_{\mathrm{st}}=\frac{\Delta Q}{\Delta u_{\mathrm{st}}}.$$

Assuming the stiffness *k*_st_ resulted from the compression of *n* cuboidal elements connected in series could be expressed as:(5)$$k_{\mathrm{st}}=\frac{k_{\mathrm{cub}}}{n},$$
where:(6)$$k_{\mathrm{cub}}=\frac{EA}{l_{\mathrm{cub}}}$$

*k*_cub_—compressive stiffness of the cuboidal element,

*E*—modulus of elasticity,

*l*_cub_—length of the cuboidal element,

*A*—cross-sectional area of the diaphragm part of the cuboidal element.

The displacement *u*_st_ of the stack model was regarded as the compressed homogeneous rod loaded with the force *Q*, without taking into account the structural solution, was equal to:(7)$$u_{\mathrm{st}}=k_{\mathrm{st}}Q.$$

The relationship (7) was also applicable when the load value *Q* was increasing or decreasing. As mentioned above, the effect of the structural solution on the system stiffness may be evaluated by referring to the real stiffness of the stack to the stiffness *k*_st_.

### 4.3. The Effect of Inaccurate Contact between Cuboidal Elements

The stiffness *k*_con_ included imperfections, that is, inaccurate contact between supporting surfaces of adjacent cuboidal elements. That value expressed the additional deformability of the stack resulting from non-dilatational deformations of elements forming the stack. The contact area between the cuboidal elements was increasing under an increasing load. It means that the stiffness *k*_con_ depended on the value of the vertical load. It was assumed by authors the change could be described with the quadratic function as follows:(8)$$k_{\mathrm{con}}\left(Q\right)=\alpha_{\mathrm{con}}Q^{2}+\beta_{\mathrm{con}}Q+\gamma_{\mathrm{con}},$$
where *α*_con_, *β*_con_, *γ*_con_ are parameters of the spring *k*_con_, determined from the tests.

Therefore, the displacement *u*_con_ was expressed with the following relationship:(9)$$u_{\mathrm{con}}=\overset{Q}{\underset{0}{\int }}\frac{dQ^{*}}{k_{\mathrm{con}}\left(Q^{*}\right)},$$
where *Q** is a dummy variable. After taking into account the boundary condition:(10)$$u_{\mathrm{con}}\left(0\right)=0$$
the following solution was obtained:(11)$$u_{\mathrm{con}}=\frac{1}{\sqrt{\Delta }}\ln\left(\frac{2\alpha_{\mathrm{con}}Q+\beta_{\mathrm{con}}-\sqrt{\Delta }}{2\alpha_{\mathrm{con}}Q+\beta_{\mathrm{con}}+\sqrt{\Delta }}\cdot \frac{\beta_{\mathrm{con}}+\sqrt{\Delta }}{\alpha_{\mathrm{con}}-\sqrt{\Delta }}\right),$$
where:(12)$$\Delta =\beta_{\mathrm{con}}^{2}-4\alpha_{\mathrm{con}}\gamma_{\mathrm{con}}.$$

The relationship (12) was also applicable when the load value *Q* was increasing or decreasing. The reason for that was the lack of the friction force on the contact surface acting in the direction of the load *Q*.

### 4.4. Effect of Initial Relative Displacements of Rolled Profiles Forming Cuboidal Elements

The relative displacement of rolled profiles, denoted by *u*_int_, occurred inside the cuboidal elements during their loading. They are generated by the imperfections in the form of initial relative displacement of rolled profiles. Displacement *u*_int_, however, can be identified with a whole group of imperfections, as a result of which there are mutual displacements of profiles inside a single rectangular element when loading the stack. The reason for those imperfections is the inaccurate assembly of rolled elements and deviations in dimensions of the profiles. The presence of those imperfections led to the additional deformability of the stack represented by the spring *k*_int_ connected in series with other stiffness components of the model. The displacements *u*_int_ are accompanied by the forces of internal friction acting in the direction of the load. Therefore, the non-elastic element connected in parallel with the spring *k*_int_ is inserted into the model. The external load *Q* was balanced by two internal forces: the elastic force *N*_int,el_ present in the spring, and the friction force *N*_int,fr_ in the non-elastic element. Values of both forces depend on the displacement *u*_int_.

On the basis of the observed relationship *Q*-Δ*l* obtained from the tests, the value of the elastic force was assumed by authors to be described with the exponential relationship:(13)$$N_{\mathrm{int},\mathrm{el}}\left(u_{\mathrm{int}}\right)=\alpha_{\mathrm{int}}\left(e^{\beta_{\mathrm{int}}u_{\mathrm{int}}}-1\right).$$

By transforming Equation (13), the relationship was obtained:(14)$$u_{\mathrm{int}}=\frac{1}{\beta_{\mathrm{int}}}\ln\left(\frac{N_{\mathrm{int},\mathrm{el}}}{\alpha_{\mathrm{int}}}+1\right).$$

The value of the internal friction *N*_int,fr_ was linearly dependent on the value of the internal elastic force. Sense of the non-conservative force of internal friction was in the opposite sense to changes of the displacement *u*_int_. Therefore, *N*_int,fr_ was the function of displacement and direction of the velocity:(15)$$N_{\mathrm{int},\mathrm{fr}}\left(u_{\mathrm{int}},{\dot{u}}_{\mathrm{int}}\right)=-\alpha_{\mathrm{fr}}N_{\mathrm{int},\mathrm{el}}\mathrm{sgn}\left({\dot{u}}_{\mathrm{int}}\right),$$
where ${\dot{u}}_{\mathrm{int}}$ is velocity of the displacement *u*_int_. The positive sense of the velocity was considered as a change of increasing value *u*_int_.

The equation of equilibrium of forces in the model of cuboidal elements takes the following form (Figure 12):(16)$$Q=N_{\mathrm{int},\mathrm{el}}+N_{\mathrm{int},\mathrm{fr}}\;\mathrm{if}\;\mathrm{sgn}\left({\dot{u}}_{\mathrm{int}}\right)>0$$
(17)$$Q=N_{\mathrm{int},\mathrm{el}}-N_{\mathrm{int},\mathrm{fr}}\;\mathrm{if}\;\mathrm{sgn}\left({\dot{u}}_{\mathrm{int}}\right)<0$$.

For increasing displacement *u*_int_, when $\mathrm{sgn}\left({\dot{u}}_{\mathrm{int}}\right)>0$ (Phase *I*) on the basis of (13), (15) and (16), the equation could be expressed as:(18)$$Q=\alpha_{\mathrm{int}}\left(e^{\beta_{\mathrm{int}}u_{\mathrm{int}}}-1\right)\left(1+\alpha_{\mathrm{fr}}\right)$$
and after transformations:(19)$$u_{\mathrm{int}}=\frac{1}{\beta_{\mathrm{int}}}\ln\left(\frac{Q}{\alpha_{\mathrm{int}}\left(1+\alpha_{\mathrm{fr}}\right)}+1\right).$$

For decreasing displacement *u*_int_, when $\mathrm{sgn}\left({\dot{u}}_{\mathrm{int}}\right)<0$ (Phase *III*) by analogy to (19), the equation can be expressed as:(20)$$Q=\alpha_{\mathrm{int}}\left(e^{\beta_{\mathrm{int}}u_{\mathrm{int}}}-1\right)\left(1-\alpha_{\mathrm{fr}}\right)$$
and:(21)$$u_{\mathrm{int}}=\frac{1}{\beta_{\mathrm{int}}}\ln\left(\frac{Q}{\alpha_{\mathrm{int}}\left(1-\alpha_{\mathrm{fr}}\right)}+1\right).$$

The model of the cuboidal element achieved the maximum displacement *u*_int,max_ when the load of the value *Q*_max_ was applied. Then, the load was reduced, which could be interpreted as applying the load Δ*Q* of the opposite sense to the force *Q*_max_ (Figure 12). At the beginning, no displacements were observed for the loading system and the velocity ${\dot{u}}_{\mathrm{int}}$ is equal to zero, which resulted from the force of static friction *N*_int,fr_. The displacement *u*_int_ occurred during unloading only if the friction force *N*_int,fr_ reached the maximum value and the sense was opposite to the velocity ${\dot{u}}_{\mathrm{int}}$. Thus, the condition inducing the displacements *u*_int_ could be expressed as:(22)$$Q_{\max}-Q-N_{\mathrm{int},\mathrm{el}}+N_{\mathrm{int},\mathrm{fr}}=0.$$

Taking into account that:(23)$$Q_{\max}=N_{\mathrm{int},\mathrm{el}}+N_{\mathrm{int},\mathrm{fr}}\;$$
a change of the load Δ*Q*, under which displacements *u*_int_ of the model is observed, is equal to:(24)$$\Delta Q=2N_{\mathrm{int},\mathrm{fr}}$$
and the value *N*_int,fr_ in Equation (24) corresponded to the displacement *u*_int,max_.

Taking into account the assumptions (15) and (23), values of internal forces can be expressed as:(25)$$N_{\mathrm{int},\mathrm{el}}=\frac{1}{1+\alpha_{fr}}Q$$
(26)$$N_{\mathrm{int},\mathrm{fr}}=\frac{\alpha_{\mathrm{fr}}}{1+\alpha_{\mathrm{fr}}}Q$$
and the value Δ*Q* can be expressed as:(27)$$\Delta Q=\frac{2\alpha_{\mathrm{fr}}}{1+\alpha_{\mathrm{fr}}}Q_{\max}.$$

Stiffness *k*_int_ determined as:(28)$$k_{\mathrm{int}}=\frac{dN_{\mathrm{int},\mathrm{el}}}{du_{\mathrm{int}}}$$
is equal to:(29)$$k_{\mathrm{int}}=\alpha_{\mathrm{int}}\beta_{\mathrm{int}}e^{\beta_{\mathrm{int}}u_{\mathrm{int}}}.$$

By substituting (14) with (29), the following is obtained:(30)$$k_{\mathrm{int}}=\beta_{\mathrm{int}}N_{\mathrm{int},\mathrm{el}}+\alpha_{\mathrm{int}}$$
which means that stiffness *k*_int_ in the analysed model was the linear function of the internal force *N*_int,el_. Characteristics of the model describing cuboidal elements are presented in Figure 12.

### 4.5. Hysteresis Loop for the Model

The assumed model was used to determine the relationship *Q*-*u*. At the beginning of the loading process (*Q* = 0), the displacement was equal to zero (*u* = 0). The increasing load *Q* induced the displacements *u*_st_, *u*_con_ and *u*_int_ (Figure 13). Greater elastic deformations of the spring *k*_int_ increased the value of the non-elastic force of friction. The force *N*_int,fr_ in that stage was the force of kinetic friction and performed work on the displacement *u*_int_. When the load *Q*_max_ was applied, the friction force *N*_fr,int_ had the greatest value. An increase in the displacement *u* caused by increasing loading was non-linear, which resulted from the relationships (2), (11) and (21) defining the displacements *u*, *u*_con_ and *u*_int_. Moreover, the model stiffness increased. The displacement reached the highest value *u*_max_ under the load *Q*_max_. That part of the relationship *Q*-*u* corresponded to Phase *I*.

In the initial phase of decreasing load *Q*, only the springs *k*_st_ and *k*_con_ deformed. The force *N*_int,fr_, being the force of kinetic friction in that stage, prevented the occurrence of the displacements *u*_int_. Displacement of the model in that phase was the sum *u* = *u*_st_ + *u*_con_. Hence, a considerable decline of the value *Q* at a relatively small change in the displacement *u* occurred. The part of the relationship *Q*-*u*, at which the load *Q* decreased by the value Δ*Q* was Phase *II* of loading, for which the load Δ*Q* was determined from (24).

When the load value was reduced by Δ*Q* = 2*N*_int,fr_(*u*_int,max_), the sense of the friction force changed. A further drop in the value *Q* was accompanied by displacements of all components of the displacement *u* and a decline in the non-elastic force *N*_int,fr_. That force, which was again the force of kinetic friction in that stage, performed work on the displacement within the range from *u*_int,max_ to 0. Because the displacements *u*_st_, *u*_con_ and *u*_int_ were revealed in the model, a drop in *Q* at the displacement change *u* was considerably smaller. The stiffness of the whole model of the stack was lower than in Phase *I*, in which three components of the displacement (*u*_st_, *u*_con_ and *u*_int_) occurred, because a greater internal force was acting during unloading of the model in the spring *k*_int_ than during loading. That was caused by relative senses of the forces *Q*, *N*_int,el_ and *N*_int,fr_ in Phase *III* (Figure 12). The value of internal forces *N*_int,fr_ and *N*_int,el_ under the load *Q* equal to zero also reached values equal to zero. Thus, the model returned to its initial state. That part of the relationship *Q*-*u* corresponded to Phase *III* observed during the tests.

## 5. Determination of the Model Parameters

The defined model of the stack was composed of four elements: one linear spring, two non-linear springs, and the element, in which the friction force acted. The linear spring was characterised by one parameter (*k*_st_). The non-linear spring *k*_con_ was characterised by three parameters (*α*_con_, *β*_con_ and *γ*_con_) and the non-linear spring *k*_int_ by two parameters (*α*_int_ and *β*_int_). The element, in which the friction force *N*_int,fr_ acted, was described by one parameter (*α*_fr_). Thus, the assumed model was characterised by seven parameters. The purpose of further analysis was to determine values of those seven parameters on the basis of results from the experimental tests discussed in Section 3.

### 5.1. Stiffness k_st_

At first, the stiffness *k*_cub_ of the cuboidal element was determined. That parameter was defined by the relationship **(6)**. For that purpose, the parts of that element were distinguished (Figure 14), in which deformations in the direction of the force *Q* were responsible for the displacement *u*_st_ (Figure 10a).

The total width of those elements equalled to 81.6 mm. The product of the width and height of the section of the cuboidal element (350 mm) was the area *A* from Equation (6). Assuming the length of the cuboidal element (*l*_cub_ = 72.5 mm) on the basis of (6), the following was determined:(31)$$k_{\mathrm{cub}}=80756\;\mathrm{MN}/m.$$

Taking into account the number of cuboidal elements, the stiffness *k*_st_ was determined on the basis of (5) as equal to:(32)$$k_{\mathrm{st}}=6211\;\mathrm{MN}/m.$$

That stiffness was the constant value, independent of the load and the eccentricity (Figure 15a). It should be emphasised that *k*_st_ was the theoretical stiffness of the stack, in which its structural solution and its imperfections were not included.

### 5.2. Stiffness k_con_

Phase *II* was taken into account to determine parameters of the spring *k*_con_ including the presence of contact area between adjacent cuboidal elements. Using the test results (graphs *Q*-Δ*l*), the relationship *Q*-*u* in that phase was approximated using the straight line (Figure 13):(33)$$Q=k_{\mathrm{II}}u+b_{\mathrm{II}}.$$

Values *k*_II_ were determined assuming the straight line (33) passed through points being the top and bottom boundary of Phase *II*. Those values were found out for all conducted tests and they are presented in Table 2. According to the assumed model, deformations in Phase *II* occurred only for two springs: *k*_con_ and *k*_st_. They were connected in series and had approximate equivalent stiffness *k*_II_, which could be expressed as:(34)$$k_{\mathrm{II}}=\frac{k_{\mathrm{st}}k_{\mathrm{con}}}{k_{\mathrm{st}}+k_{\mathrm{con}}}.$$

Because the stiffness *k*_st_ was known, the relationship (34) was the base to determine the stiffness *k*_con_ for all conducted tests. Approximate stiffness values *k*_con_ calculated for a given test are shown in Table 2. That table also contains relationships for the stiffness *k*_con_ as a function of the loading force for a given value of the eccentricity *e*_a_. The relationship was determined using the method of least squares. The obtained parameters *α*_con_, *β*_con_, and *γ*_con_ achieved satisfactory compatibility—the value of determination coefficient *R*^2^ was not smaller than 0.991. The relationships describing the stiffness *k*_con_ corresponding to particular eccentricities are illustrated in the graph (Figure 15b). The analysis of the relationship indicated the reduction of the spring stiffness *k*_con_ with increasing values of the eccentricity. The graph in Figure 15b presents curves for the range of loads applied during the tests. Therefore, the exemplary range of the curve for *e*_a_ = 40 mm corresponded to values *Q* within the range from 0 to 100 kN.

### 5.3. Stiffness k_int_ and Force N_int,fr_

Determination of the stiffness *k*_int_ as a function of the load *Q* required the defined parameters *α*_int_, *β*_int_ and *α*_fr_. To find the parameters *α*_int_ and *β*_int_, the graph of the relationship *N*_int,el_-*u*_int_, was determined from the tests. It was the mean of the relationships *Q*-*u*_int_ between Phase *I* and Phase *III* (Figure 12). Then, the parameters *α*_int_ and *β*_int_ of the approximation in the form of (13) were determined using the method of least squares. Similarly, the tested relationship *N*_int,fr_-*u*_int_, making half of the difference in the relationship *N*_int,el_-*u*_int_, between Phase *I* and Phase *III* was determined to obtain the parameter *α*_fr_ (Figure 12). Then, the parameter *α*_fr_ of the approximation in the form of (15) was determined using the method of least squares.

The procedures described above were applied to all tested eccentricities *e*_a_, and obtained parameters and functions describing the stiffness *k*_int_ and *N*_int,fr_ for individual eccentricities are presented in Table 3 and in Figure 16.

## 6. Analysis of the Model

The defined model confirmed the presence of three phases in the loading-unloading process for the model. Each phase displayed linear-elastic properties resulting from the section stiffness of cuboidal elements and non-linear elastic properties and friction representing the effect of geometrical imperfections. The model was characterised by seven parameters, the values of which were determined as described in Section 5. Elastic and non-elastic properties of the stack depended on the load values and the phase of the loading–unloading process.

Graphs were prepared to compare results obtained from the model and test results as illustrated in Figure 17. The results for *Q*-Δ*l* obtained from testing the relationship *Q*-*u* and from the model were included in the graph. The relationships used to prepare graphs *Q*-*u* are compared in Table 4. The lines *Q*-*u* (the model) and *Q*-Δ*l* (the tests) were practically overlapping. Consequently, the assumed model seemed to describe effects observed in the stack satisfactorily.

The impact of imperfections on the stack stiffness was analysed by testing the characteristics of springs being components of the model. The graphs in Figure 18 demonstrate the relationships *k*_st_(*Q*), *k*_con_(*Q*) and *k*_int_(*Q*). Each subsequent graph corresponds to the stacks loaded at individual eccentricities. Values of the model properties calculated for selected load values are presented in Table 5. The stiffness *k*_st_ was the stack stiffness resulting from the section of cuboidal elements taking part in transferring the force *Q* in the longitudinal direction. That stiffness value was constant and equal to 6211 MN/m. It was independent of the eccentricity of the applied force and its value. As described above, it was also the theoretical stiffness that could be characteristic for the stack, in which deformations related to the structural solutions and imperfections did not occur.

Under the load *Q* equal to zero, the stiffness *k*_con_ was 20 MN/m. An increase in loading caused an increase in the value *k*_con_ described by the quadratic function. The value *k*_con_ was smaller for the stacks with greater eccentricity. Under the load *Q* equal to 500 kN, *k*_con_ was 2498 MN/m at the eccentricity *e*_a_ equal to zero, and at the eccentricity *e*_a_ equal to 30 mm, the stiffness was 1273 MN/m (Table 5).

The stiffness *k*_int_ corresponded to the internal stiffness of the cuboidal element related to relative displacements of rolled profiles forming a given cuboidal element, and the deformability of welds. In the assumed model, the stiffness was described by the linear function and its value was increasing as the internal force *N*_int,el_ increased. Under the load *Q* equal to zero, the stiffness value ranged from 9 to 16 MN/m depending on the eccentricity. An increasing load caused an increase in the stiffness value. Higher values corresponded to greater eccentricities as well. Under the load *Q* equal to 500 kN, the stiffness was 596 MN/m at the eccentricity *e*_a_ equal to zero, and at the eccentricity *e*_a_ equal to 30 mm the stiffness was 1142 MN/m (Table 5).

The spring *k*_st_, *k*_con_ and *k*_int_ making the model were connected in series. Thus, the term of the equivalent stiffness could be introduced with reference to the springs forming the model.

The equivalent stiffness of stack affected by inaccurate contact may be expressed by the formula:(35)$$k_{\mathrm{st},\mathrm{con}}=\frac{k_{\mathrm{st}}k_{\mathrm{con}}}{k_{\mathrm{st}}+k_{\mathrm{con}}}.$$

Similarly, the equivalent stiffness of stack affected by initial relative displacements of rolled profiles inside the cuboidal element is given by the expression:(36)$$k_{\mathrm{st},\mathrm{int}}=\frac{k_{\mathrm{st}}k_{\mathrm{int}}}{k_{\mathrm{st}}+k_{\mathrm{int}}}.$$

All three springs in the model in Phases *I* and *III* were exposed to deformations. Therefore, the equivalent stiffness *k*_I,III_ was defined and determined from the following relationship:(37)$$k_{I,\mathrm{III}}=\frac{k_{\mathrm{st}}k_{\mathrm{con}}k_{\mathrm{int}}}{k_{\mathrm{st}}k_{\mathrm{con}}+k_{\mathrm{st}}k_{\mathrm{int}}+k_{\mathrm{con}}k_{\mathrm{int}}}.$$

In Phase *II* only the springs *k*_st_, *k*_con_, in the model were deformed and thus the equivalent stiffness *k*_II_ is equal to *k*_st,con_:(38)$$k_{\mathrm{II}}=k_{\mathrm{st},\mathrm{con}}.$$

Values of the equivalent stiffness *k*_I,III_ were also plotted on graphs in Figure 18 and their comparison for selected eccentricity values is in Table 5. Moreover, those stiffness values are illustrated for all eccentricities in Figure 19a. The equivalent stiffness *k*_I,III_ for the support model was increasing almost linearly with the increase in load. Under the load equal to zero, the stiffness value ranged from 2 to 9 MN/m depending on the eccentricity. Under the load equal to 500 kN, the stiffness value ranged from 447 MN/m at the eccentricity *e*_a_ = 0 to 549 MN/m at *e*_a_ = 30. Thus, the impact of the eccentricity of the equivalent stiffness of the stack was rather negligible.

The conducted analysis demonstrated the structural solutions and resulting geometrical imperfections had a significant effect on the stack stiffness. If the stack stiffness depended only on the section of cuboidal elements and the stack length, then its stiffness would be equal to *k*_st_. The equivalent stiffness *k*_I,III_ covered the impact of all factors included in the model. Therefore, the effect of the structural solution on the stack stiffness could be expressed by the ratio *k*_I,III_/*k*_st_. The graph representing the ratio for all stack models is shown in Figure 19b. Its value changed almost linearly in a function of the load *Q*. For *Q* = 0, its value was close to zero, and for *Q*_max_ = 1000 kN the value was equal to 0.15. It means that the stack stiffness was dominated by imperfections.

A change in the stiffness *k*_II_ = *k*_st,con_ corresponding to individual eccentricities was plotted on adequate graphs in Figure 18 and collectively for all eccentricities in Figure 20a. The equivalent stiffness values in Phase *II* were ca. four times greater than the equivalent stiffness values *k*_I,III_. They varied from 1056 to 1781 MN/m under the load *Q* equal to 500 kN. Higher values *k*_II_ were dominated by the constant *k*_con_, characterised by higher values at lower eccentricities. The effect of geometrical imperfections of inaccurate contact between the stack elements (*k*_con_) on the stack stiffness in Phase *II* can be described by the ratio *k*_st,con_/*k*_st_. The graph representing the ratio for all stack models is shown in Figure 20b. The change of its value was almost linear in a function of the load *Q*. For *Q* = 0, its value was close to zero, and for *Q*_max_ = 1000 kN the value was equal to 0.55.

A change in the stiffness *k*_st,int_ for all eccentricities is plotted in Figure 21a. The values of equivalent stiffness *k*_st,int_ are almost equal to the values of equivalent stiffness *k*_I,III_ for eccentricities greater than 20 mm. The effect of the imperfection of inaccurate contact between the stack elements (*k*_int_) on the equivalent stack stiffness *k*_st,int_ can be described by the ratio *k*_st,int_/*k*_st_. The ratio for all stack models is shown in Figure 21b. Similarly, its value changed almost linearly in a function of the load *Q*. For *Q* = 0, its value was close to zero, and for *Q*_max_ = 1000 kN the value was equal to 0.16.

The analysis of equivalent stiffness values *k*_I,III_ and *k*_II_ unambiguously indicated the stack stiffness was predominantly affected by the imperfection related to initial relative displacements of rolled profiles inside the cuboidal element. As a consequence, internal non-elastic forces were observed. Those forces also represented the structural friction between displacing rolled profiles that formed cuboidal elements. The presence of those forces resulted in the formation of hysteresis loops in the model. Values of those forces linearly depended on the load value and amounted to ca. 10–12% of the applied load at the eccentricity *e*_a_ ≤ 30 mm, and 1.7% at the eccentricity *e*_a_ = 40 mm. The eccentricity of 30 mm or smaller had a slight effect on the non-elastic internal force in the stack.

## 7. Conclusions

The stacks of cuboidal elements used in, among others, rectification processes of vertically deflected buildings, are loaded eccentrically and have geometrical imperfections. The eccentric load of the stack results from the inaccurate assembly of the jack on the stack. The imperfections result from the structural solution. Two types of imperfections were distinguished. The first type included the inaccurate contact between the cuboidal elements, and the second one referred to initial relative displacements of profiles inside the cuboidal elements. Those imperfections were the consequences resulting from deviations in dimensions of rolled profiles and the inaccuracy of relative position of profiles during welding.

The tests on the equivalent system were conducted under laboratory conditions to represent the behaviour of a 1 m long stack that was the component of the building support. Cyclic loading of the stack was made at the test stand. The force was applied with the eccentricity equal to 0, 10, 20, 30 and 40 mm. The load increased from 1.5 to 1000 kN. Deformations of the material were found to be elastic, whereas the relationships between the force and the change in the stack length were non-linear. Moreover, the hysteresis loops were observed. However, the system returned to its initial position after its unloading. It meant that apart from elastic forces, internal forces of the friction acted in the stack. They were responsible for activating or deactivating relative elastic displacements of rolled profiles that formed the cuboidal elements. At the beginning of loading (1.5 kN), the stack stiffness, depending on the eccentricity, ranged from 2 to 9 MN/m, under the load of 500 kN from 447 to 549 MN/m, and under the load of 1000 kN from 881 to 946 MN/m. Slightly higher values of the stiffness were obtained for greater eccentricities.

The mathematical model of the stack representing its deformations during the load-unload cycle within the range of 1.5–1000 kN was composed of four elements: one linear spring, two non-linear springs, and the element, in which the friction force acted. The linear spring represented the stiffness of the stack regarded as the rod of the equivalent section. The first non-linear spring described by an increasing quadratic function in a function of loading represented displacements corresponding to the imperfections of inaccurate contact between the cuboidal elements. The second non-linear spring described by an increasing linear function modelled the displacements corresponding to the imperfections of initial relative displacements of rolled profiles that formed the cuboidal elements. The springs in the defined model were connected in series, and the element, in which the forces of static friction were observed, was connected in parallel with the second non-linear spring. The model was described with seven parameters, in which the values were determined on the basis of the tests.

Three springs were displaced during loading of the model (Phase *I*). At the beginning of unloading (Phase *II*), the friction forces prevented displacements in the second non-linear spring. Hence, the model had a four times greater stiffness than in the final phase of unloading when three springs were again deformed (Phase *III*).

The analysis of the model demonstrated the eccentricity values in Phases *I* and *III* slightly increased the stack stiffness by ca. 4–10% when increasing the eccentricity by 10 mm. The eccentricity increase in Phase *II* by 10 mm caused a drop in the stack stiffness by ca. 11–22%.

The presence of imperfections in the stack considerably decreased its stiffness. However, that effect was decreasing as the load was increasing. Stiffness of the stack with a length of 1 m, regarded as the system without any imperfections, was equal to 6211 MN/m. The imperfections of initial relative displacements of rolled profiles forming the cuboidal elements had the greatest impact on the stack stiffness. For example, the imperfections caused a 10-fold drop in the stack stiffness under the load of 500 kN, and the stiffness dropped by ca. 3.5 times when the imperfections consisted in the inaccurate contact between rolled profiles. Finally, the combined imperfections caused almost a 14-fold drop in stiffness of the stack with a length of 1 m under the load of 500 kN from 6212 to ca. 450 MN/m.

## Figures and Tables

**Figure 1 materials-13-03174-f001:**
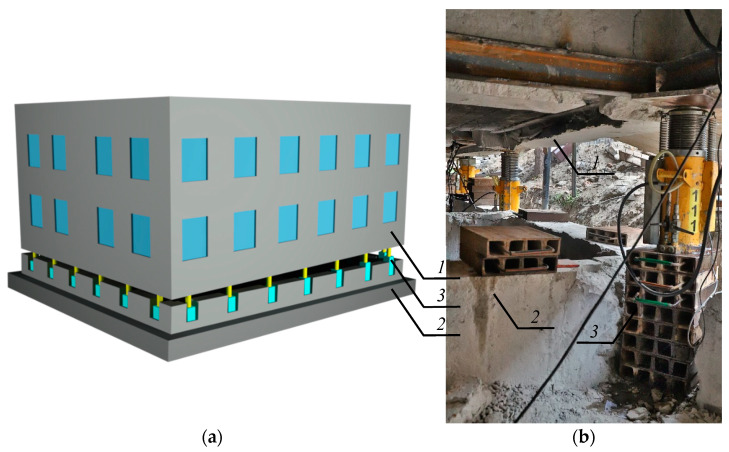
Exemplary use of a stack of steel elements during removal of the building deflection, (**a**) principle of removing deflection by uneven lifting of the building, (**b**) the stack installed in the building wall; *1*—unevenly lifted part of the building, *2*—part remaining in the ground, *3*—the stack of cuboidal elements.

**Figure 2 materials-13-03174-f002:**
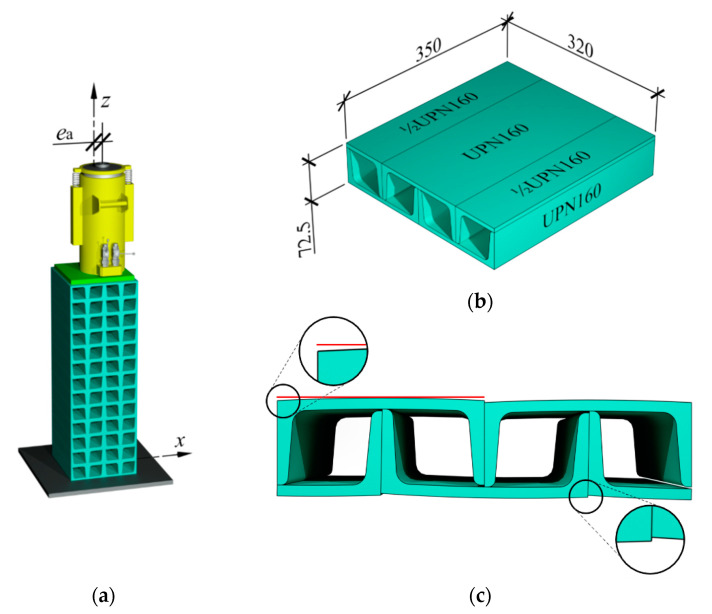
Temporary support: (**a**) elements being components of support, (**b**) cuboidal element, (**c**) geometrical imperfections in the form of inaccurate contact between cuboidal elements and initial displacements of rolled profiles.

**Figure 3 materials-13-03174-f003:**
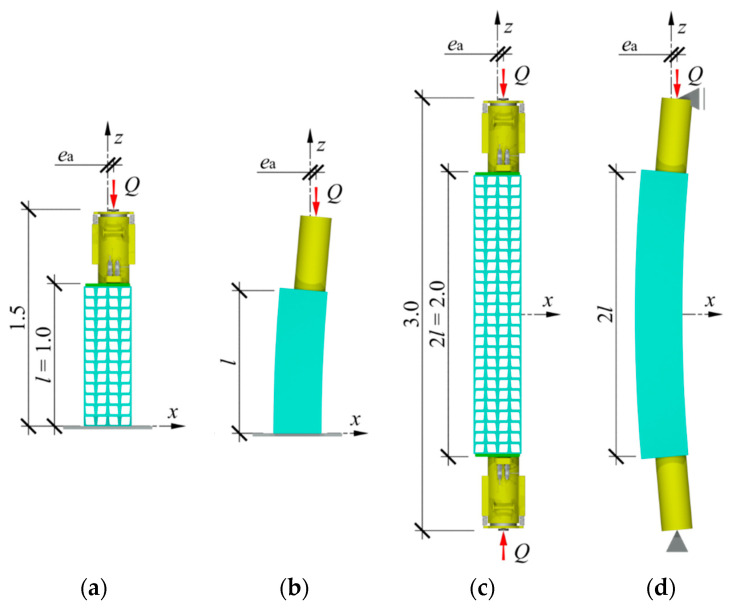
Temporary support with eccentricity *e*_a_ and tested system: (**a**) location of support elements in relation to each other, (**b**) deformation of support, (**c**) tested system, (**d**) deformation of tested system.

**Figure 4 materials-13-03174-f004:**
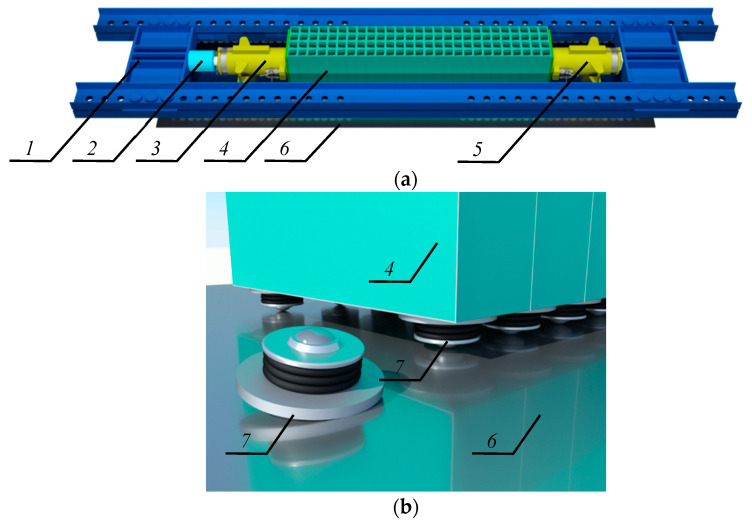
Laboratory stand: (**a**) general view, (**b**) ball bearings; *1*—steel frame, *2*—compression force transducer, *3*—active jack, *4*—stack of steel cuboidal elements, *5*—passive jack, *6*—platform made of polished steel plate, *7*—ball transfer unit.

**Figure 5 materials-13-03174-f005:**
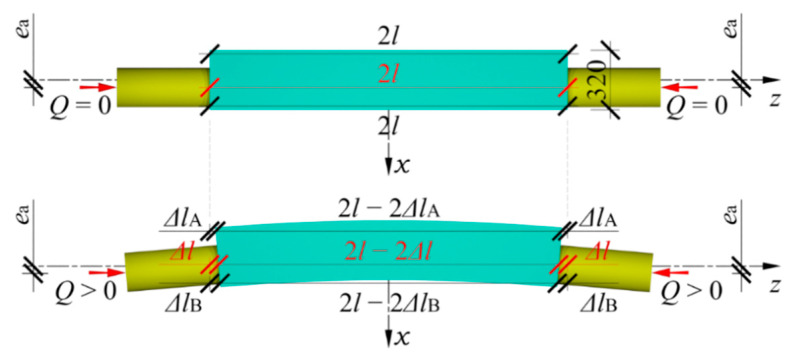
Change in length Δ*l*_A_, Δ*l*_B_ of the stack edge and in stack length Δ*l*.

**Figure 6 materials-13-03174-f006:**
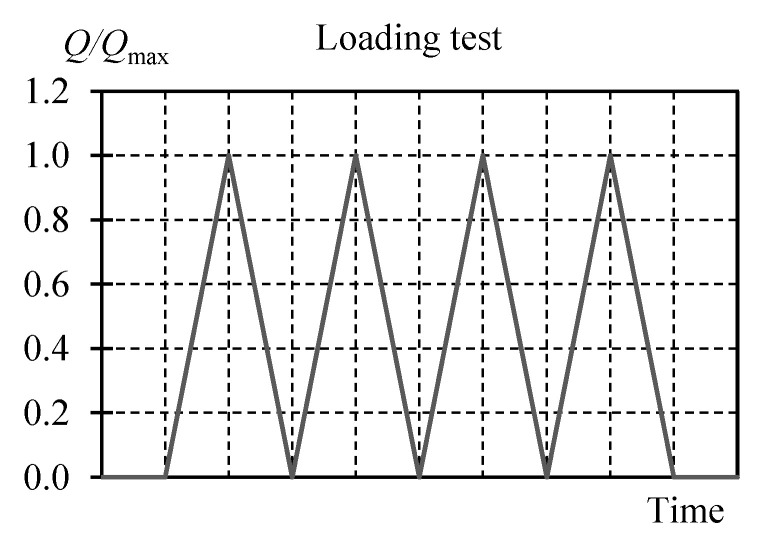
Load *Q* change during tests.

**Figure 7 materials-13-03174-f007:**
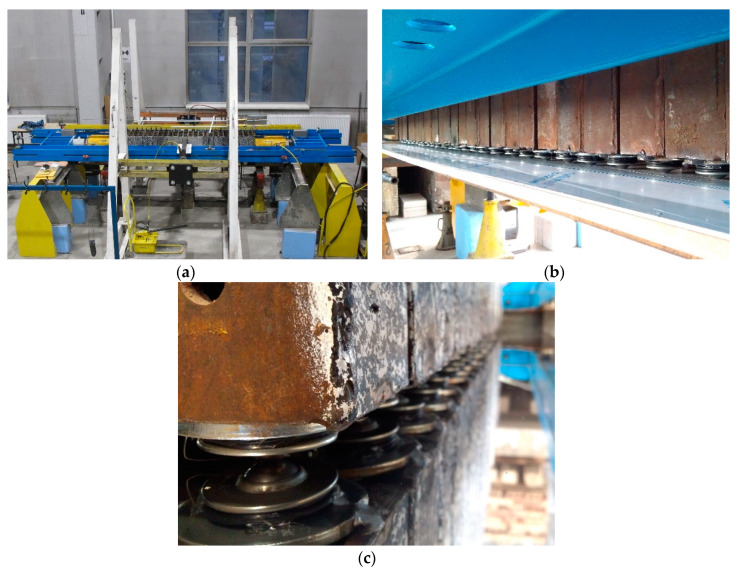
Test stand: (**a**) general view, (**b**,**c**) ball transfer units and platform made of polished steel plate.

**Figure 8 materials-13-03174-f008:**
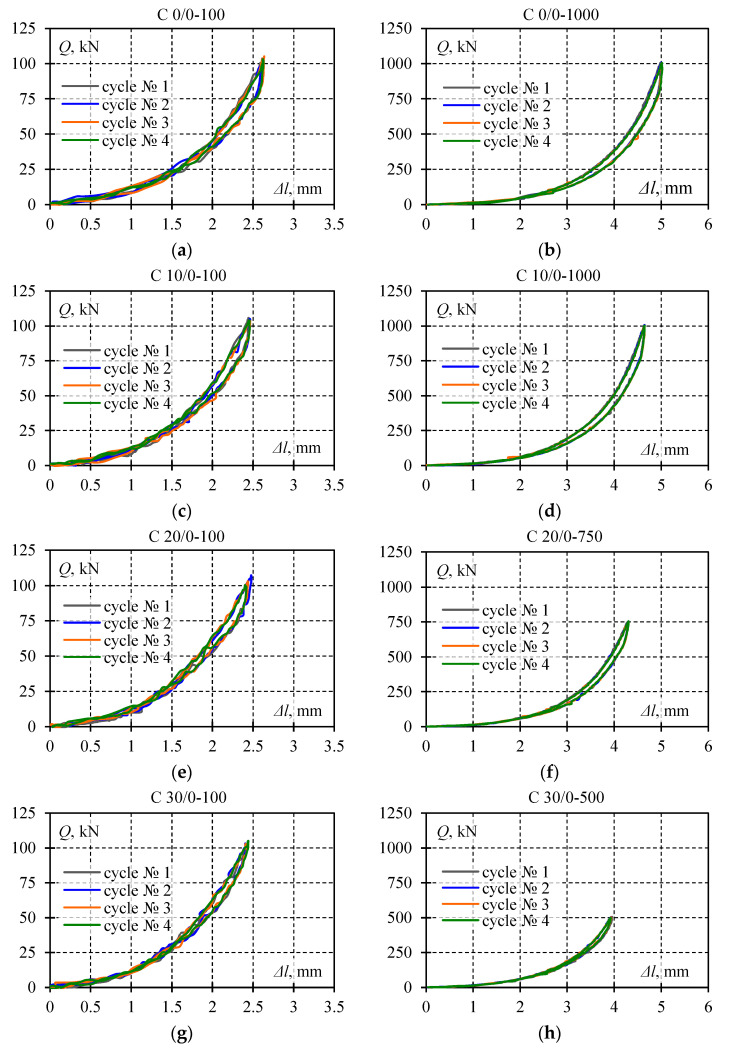
Measurement results corresponding to maximum and minimum values *Q*_max_ for an eccentricity value *e*_a_; each subsequent diagram illustrates the following tests (Table 1): (**a**) C 0/0-100, (**b**) C 0/0-1000, (**c**) C 10/0-100, (**d**) C 10/0-1000, (**e**) C 20/0-100, (**f**) C 20/0-750, (**g**) C 30/0-100, (**h**) C 30/0-500, (**i**) C 40/0, (**j**) test procedure and identification.

**Figure 9 materials-13-03174-f009:**
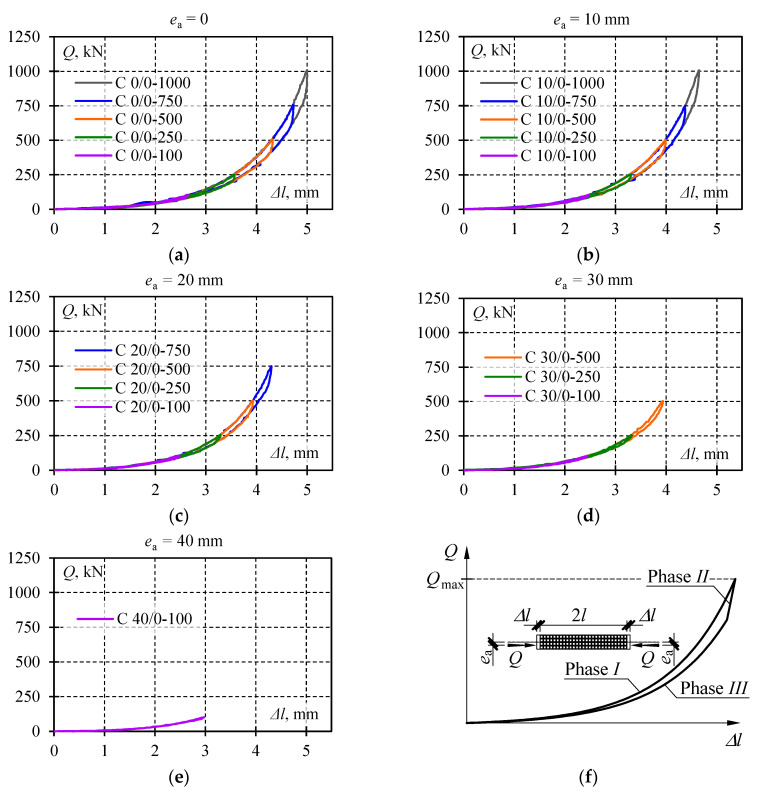
Measurement results for *Q*-Δ*l* at the following eccentricities: (**a**) *e*_a_ = 0, (**b**) *e*_a_ = 10 mm, (**c**) *e*_a_ = 20 mm, (**d**) *e*_a_ = 30 mm, (**e**) *e*_a_ = 40 mm; (**f**) phases *I*, *II*, *III* identified for each *Q*-Δ*l* cycle.

**Figure 10 materials-13-03174-f010:**
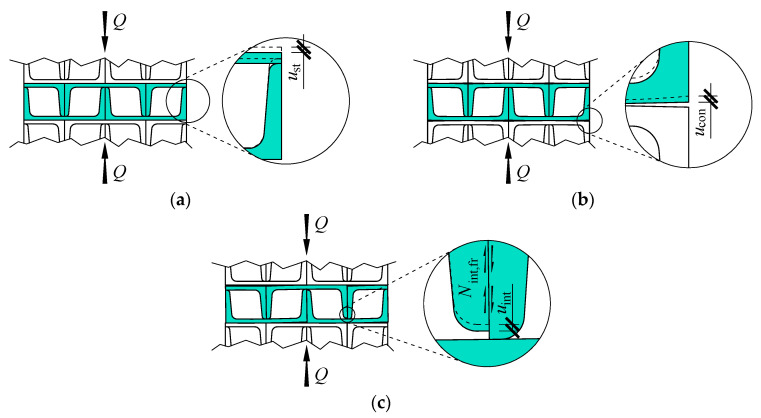
Components of the stack displacement: (**a**) *u*_st,*i*_, (**b**) *u*_con,*i*_, (**c**) *u*_int,*i*_.

**Figure 11 materials-13-03174-f011:**
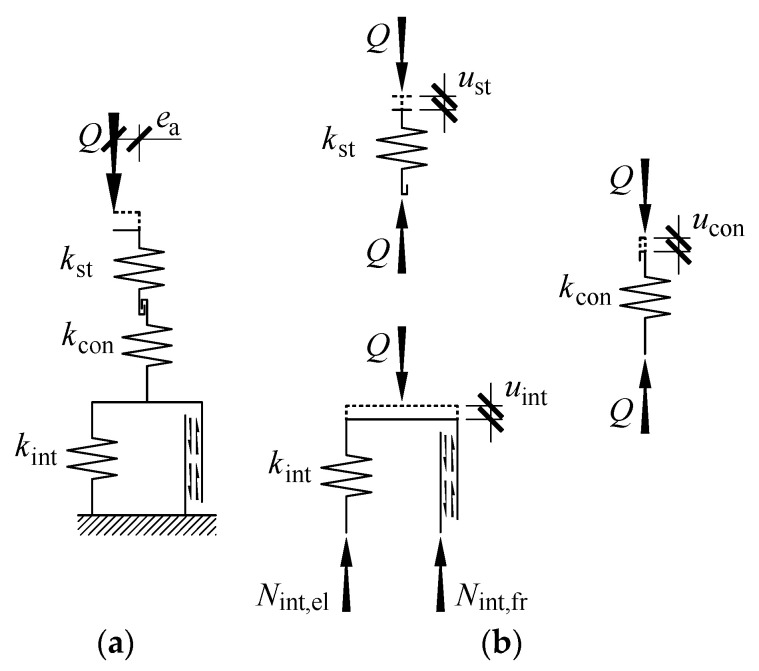
Model of the support: (**a**) sketch of the model (**b**) displacements of the model elements and forces appearing in them.

**Figure 12 materials-13-03174-f012:**
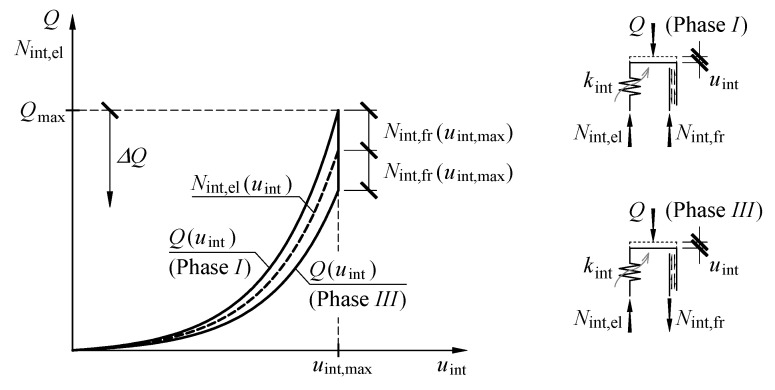
Principle of determining the displacement *u*_int_ in the model.

**Figure 13 materials-13-03174-f013:**
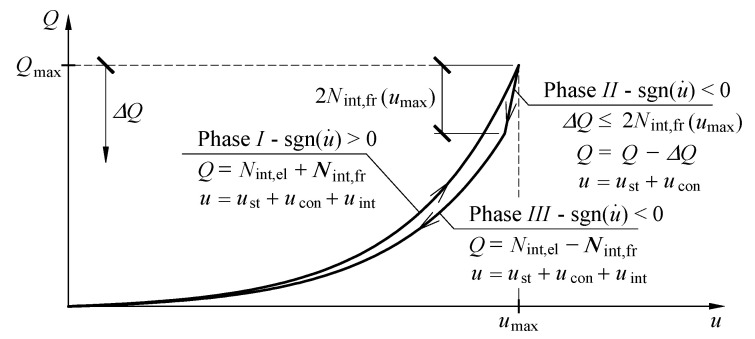
Displacements of the model under an increasing (Phase *I*) and decreasing load (Phase *III*, Phase *II*).

**Figure 14 materials-13-03174-f014:**
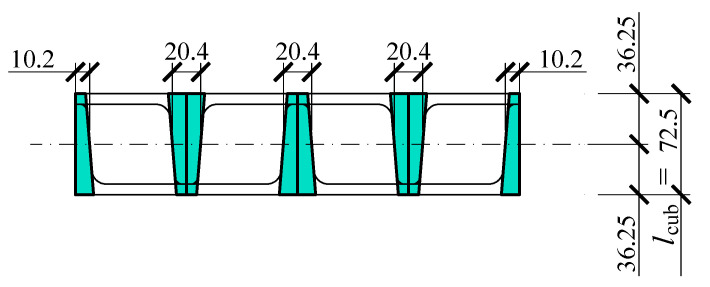
Parts of the cuboidal element in which the deformations in the direction of the force *Q* were responsible for the displacement *u*_st_ (the total width of interacting elements was equal to 2 × 10.2 + 3 × 20.4 = 81.6 mm).

**Figure 15 materials-13-03174-f015:**
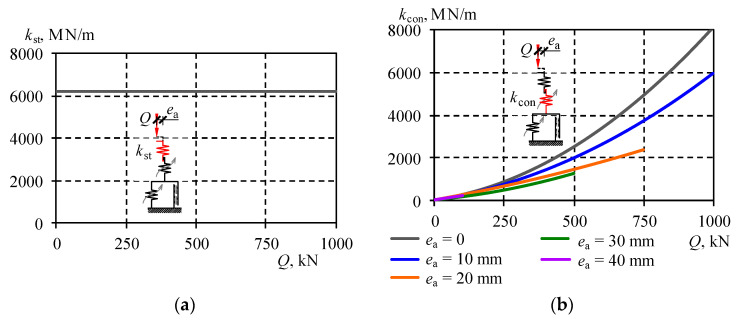
Determined stiffness values of the model: (**a**) *k*_st_, (**b**) *k*_con_.

**Figure 16 materials-13-03174-f016:**
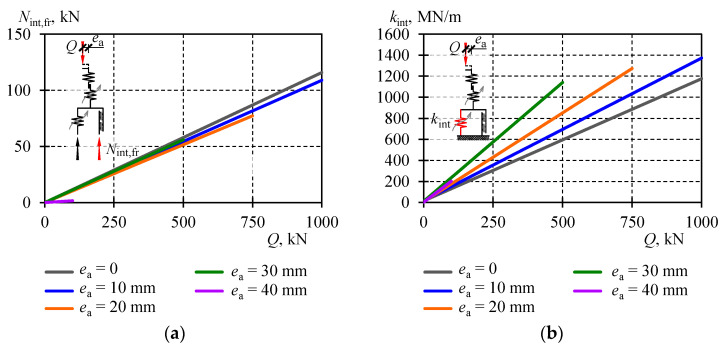
(**a**) Internal friction force *N*_int,fr_, (**b**) stiffness component of the model *k*_int_.

**Figure 17 materials-13-03174-f017:**
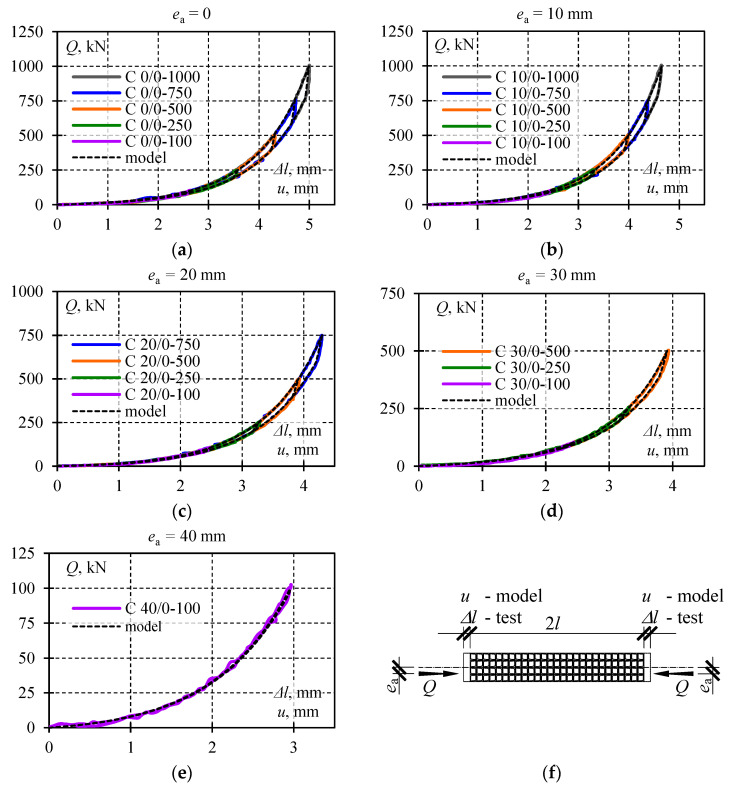
Compared hysteresis loops obtained from the model (lines *Q*-*u*) with loops determined from tests (lines *Q*-Δ*l*) for the following eccentricities: (**a**) *e*_a_ = 0, (**b**) *e*_a_ = 10 mm, (**c**) *e*_a_ = 20 mm, (**d**) *e*_a_ = 30 mm, (**e**) *e*_a_ = 40 mm, (**f**) testing scheme and denotations.

**Figure 18 materials-13-03174-f018:**
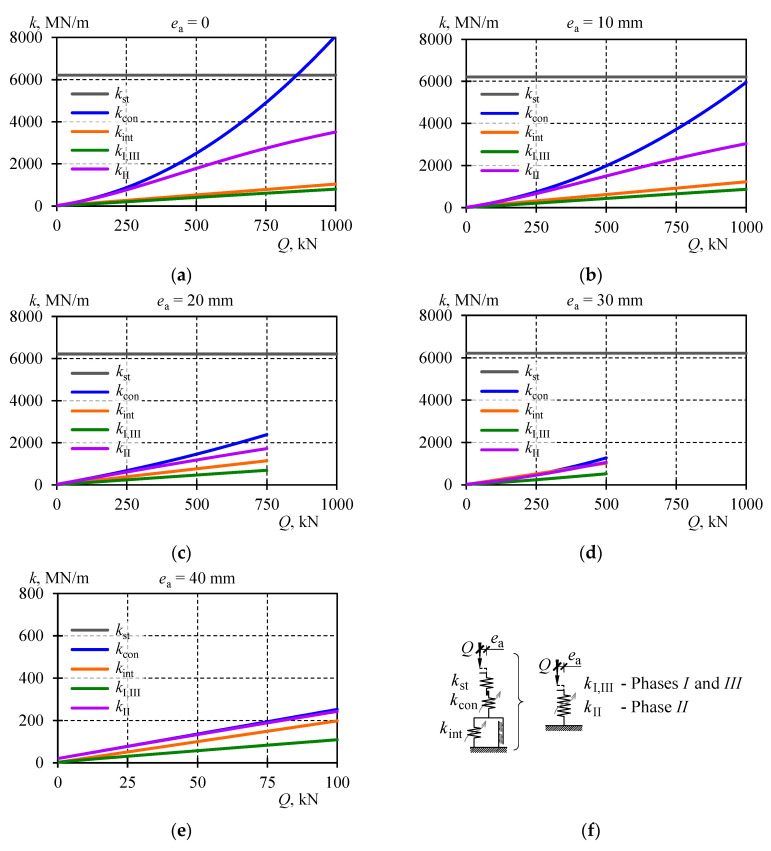
Components *k*_st_, *k*_cont_, *k*_int_ of the stiffness for the stack model and equivalent stiffness *k*_I,III_ by (37) and *k*_II_ by (33) corresponded to the following eccentricities: (**a**) *e*_a_ = 0, (**b**) *e*_a_ = 10 mm, (**c**) *e*_a_ = 20 mm, (**d**) *e*_a_ = 30 mm, (**e**), *e*_a_ = 40 mm, (**f**) denotations of springs.

**Figure 19 materials-13-03174-f019:**
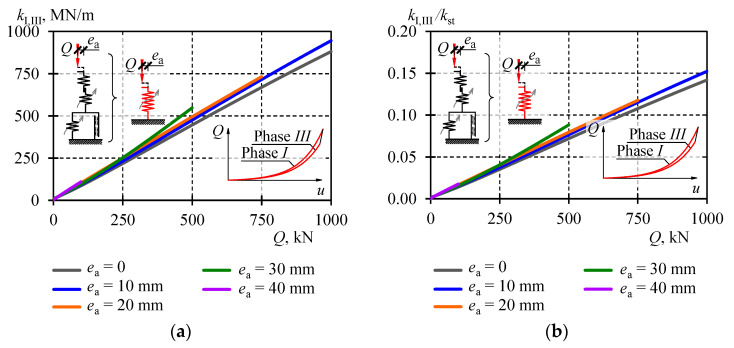
Equivalent stiffness *k*_I,III_ in Phase *I* and Phase *III*: (**a**) equivalent stiffness *k*_I,III_, (**b**) ratio *k*_I,III_*/k*_st_.

**Figure 20 materials-13-03174-f020:**
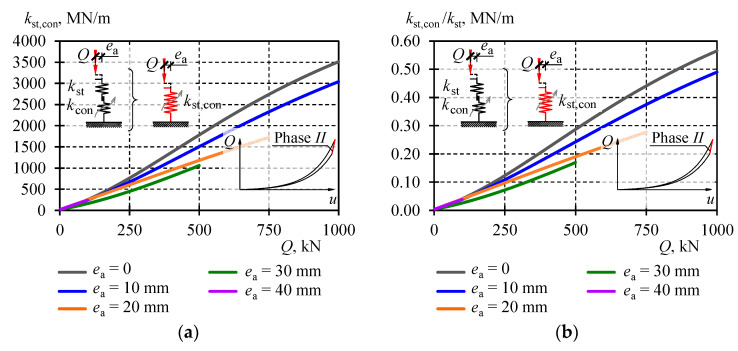
Equivalent stiffness *k*_st,con_ in Phase *II*: (**a**) equivalent stiffness *k*_st,con_, (**b**) ratio *k*_st,con_*/k*_st_.

**Figure 21 materials-13-03174-f021:**
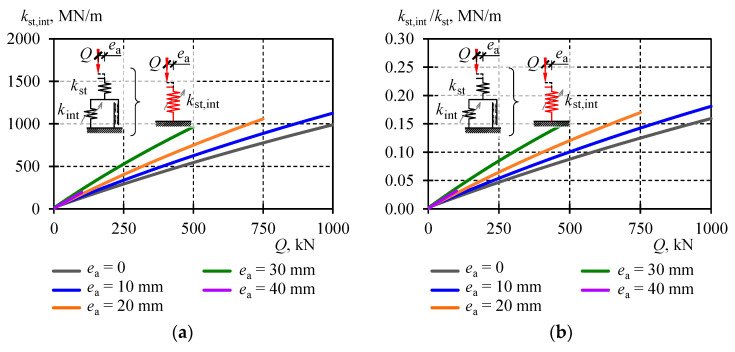
Equivalent stiffness *k*_st,int_: (**a**) equivalent stiffness *k*_st,int_, (**b**) ratio *k*_st,int_*/k*_st_.

**Table 1 materials-13-03174-t001:** Comparison of tests.

Test Name	*e*_a_ [mm]	*Q*_min_ [kN]	*Q*_max_ [kN]	Number of Loops
C 0/0-100	0	0	100	4
C 0/0-250	0	0	250	4
C 0/0-500	0	0	500	4
C 0/0-750	0	0	750	4
C 0/0-1000	0	0	1000	4
C 10/0-100	10	0	100	4
C 10/0-250	10	0	250	4
C 10/0-500	10	0	500	4
C 10/0-750	10	0	750	4
C 10/0-1000	10	0	1000	4
C 20/0-100	20	0	100	4
C 20/0-250	20	0	250	4
C 20/0-500	20	0	500	4
C 20/0-750	20	0	750	4
C 30/0-100	30	0	100	4
C 30/0-250	30	0	250	4
C 30/0-500	30	0	500	4
C 40/0-100	40	0	100	4

**Table 2 materials-13-03174-t002:** Stiffness values obtained from tests and the function describing the characteristics of the spring *k*_con_.

Test Name	*Q*_max_ [kN]	*k*_II_ [MN/m]	*k*_con_ [MN/m]	*k*_con_(*Q*) = *α*_con_*Q*^2^ + *β*_con_*Q* + *γ*_con_ [MN/M]	*R* ^2^
C 0/0-100	100	501	545	0.0062(*Q*)^2^ + 1.856(*Q*) + 20	0.995
C 0/0-250	250	921	1082		
C 0/0-500	500	1761	2458		
C 0/0-750	750	2696	4764		
C 0/0-1000	1000	3556	8317		
C 10/0-100	100	347	367	0.0040(*Q*)^2^ + 1.934(*Q*) + 20	0.995
C 10/0-250	250	711	803		
C 10/0-500	500	1386	1784		
C 10/0-750	750	2416	3953		
C 10/0-1000	1000	3035	5937		
C 20/0-100	100	290	304	0.0011(*Q*)^2^ + 2.325(*Q*) + 20	0.998
C 20/0-250	250	568	625		
C 20/0-500	500	1211	1504		
C 20/0-750	750	1722	2383		
C 30/0-100	100	227	236	0.0026(*Q*)^2^ + 1.206(*Q*) + 20	0.991
C 30/0-250	250	428	460		
C 30/0-500	500	1065	1285		
C 40/0-100	100	240	250	0.0001(*Q*)^2^ + 2.315(*Q*) + 20	1.000

**Table 3 materials-13-03174-t003:** Experimentally determined coefficients *α*_int_, *β*_int_, *α*_fr_ and functions describing stiffness *k*_int_ and friction *N*_int,fr_ under the load *Q*.

Eccentricity *e*_a_	*α*_int_ [-]	*β*_int_ [-]	*α*_fr_ [-]	*k*_int_ [MN/m]	*N*_int,fr_ [kN]
0	15.66	1.313	0.131	1.161(*Q*) + 15.66	0.116(*Q*)
10 mm	15.987	1.524	0.122	1.358(*Q*) + 15.987	0.109(*Q*)
20 mm	7.815	1.881	0.115	1.687(*Q*) + 7.815	0.103(*Q*)
30 mm	11.033	2.546	0.126	2.261(*Q*) + 11.033	0.112(*Q*)
40 mm	2.485	2.026	0.017	1.992(*Q*) + 2.485	0.017(*Q*)

**Table 4 materials-13-03174-t004:** Relationships describing phases of the model.

Phase	Range of Load *Q*	*u* _st_	*u* _con_	*u* _int_	*u*
		(acc. to Relationship)			
*I*	0-$Q_{\max}$	(7)	(11)	(19)	*u*_st_ + *u*_con_ + *u*_int_
*II*	$Q_{\max}$-$Q_{\max}\frac{2\alpha_{\mathrm{fr}}}{1+\alpha_{\mathrm{fr}}}$			-	*u*(*Q*_max_) − *u*_st_(Δ*Q*) − *u*_con_(Δ*Q*)
*III*	$Q_{\max}\frac{2\alpha_{\mathrm{fr}}}{1+\alpha_{\mathrm{fr}}}$-0			(21)	*u*_st_ + *u*_con_ + *u*_int_

**Table 5 materials-13-03174-t005:** Values of stiffness and internal forces in the model elements.

Eccentricity	Property	*Q* = 0 kN	*Q* = 100 kN	*Q* = 250 kN	*Q* = 500 kN	*Q* = 750 kN	*Q* = 1000 kN
*e*_a_ = 0–40 mm	*k*_st_, MN/m	6211	6211	6211	6211	6211	6211
*e*_a_ = 0 mm	*k*_con_, MN/m	20	268	872	2498	4900	8076
	*k*_int_, MN/m	16	132	306	596	886	1177
	*k*_I,III_, MN/m	9	87	218	447	670	881
	*k*_II_, MN/m	20	257	764	1781	2739	3511
	*N*_int,fr_, kN	0	12	29	58	87	116
	*N*_int,el_, kN	0	88	221	442	663	884
*e*_a_ = 10 mm	*k*_con_, MN/m	20	253	754	1987	3721	5954
	*k*_int_, MN/m	16	152	355	695	1034	1374
	*k*_I,III_, MN/m	9	93	232	475	716	946
	*k*_II_, MN/m	20	243	672	1505	2327	3040
	*N*_int,fr_, kN	0	21	53	105	158	210
	*N*_int,el_, kN	0	79	198	395	593	790
*e*_a_ = 20 mm	*k*_con_, MN/m	20	264	670	1458	2383	-
	*k*_int_, MN/m	8	177	430	851	1273	-
	*k*_I,III_, MN/m	6	104	251	495	732	-
	*k*_II_, MN/m	20	253	605	1180	1722	-
	*N*_int,fr_, kN	0	19	47	94	140	-
	*N*_int,el_, kN	0	81	203	407	610	-
*e*_a_ = 30 mm	*k*_con_, MN/m	20	167	484	1273	-	-
	*k*_int_, MN/m	11	237	576	1142	-	-
	*k*_I,III_, MN/m	7	96	252	549	-	-
	*k*_II_, MN/m	20	162	449	1056	-	-
	*N*_int,fr_, kN	0	18	44	88	-	-
	*N*_int,el_, kN	0	83	206	413	-	-
*e*_a_ = 40 mm	*k*_con_, MN/m	20	253	-	-	-	-
	*k*_int_, MN/m	2	202	-	-	-	-
	*k*_I,III_, MN/m	2	110	-	-	-	-
	*k*_II_, MN/m	20	243	-	-	-	-
	*N*_int,fr_, kN	0	13	-	-	-	-
	*N*_int,el_, kN	0	87	-	-	-	-

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
