# Peer review of "Effect of Eccentricity of Applied Force and Geometrical Imperfections on Stiffness of Stack of Cuboidal Steel Elements"

_materials, 2020, doi:10.3390/ma13143174_

Round 1

Reviewer 1 Report

Effect of Eccentricity of Applied Force and Geometrical Imperfections on Stiffness of Stack of Cuboidal Steel Elements (review)

The paper deals with a stack of cuboidal steel elements and addresses a problem of the effect of external force eccentricity on stiffness. Such stacks are temporary supports in building rectification, which utilizes hydraulic piston jacks in walls. Their inaccurate position causes eccentricity of loading. Such eccentricity and especially two imperfections, i.e. initial relative displacements of rolled profiles inside cuboidal elements and inaccurate contact between them, significantly reduce stiffness. The authors performed experimental tests, developed a mathematical model, and calibrated it to the measurements. The paper is well written with a systematic and logical flow and is mathematically consistent. From a mathematical and scientific point of view, it is not particularly complex, but the content is interesting and ideas about how to tackle the problem, are innovative, thus it deserves a publication after the following dilemmas will be clarified and after a revision will be made.

The specific remarks are as follows:

  1. Line 134: The system was installed in the horizontal position. This neglects the influence of the dead weight of rolled profiles. Namely, in a vertical position, such a stack is axially loaded with dead weight as a body force. Please, comment on the neglect of this effect, especially at low values of the loading force.
  2. Figure 4: How did you measure displacement and how did you exclude the influence of frame deformations? Please, add the comment about the importance of the frame rigidity.
  3. Line 272: There is a strange definition for A ("area... acting at transferring"?). In eq. 6 it represents the area of cross-section, but it is difficult to define it at that point. Afterward, it is correctly used, but try to define it better.
  4. Table 3 and Figure 16a: Why is the slope of ea=40mm so low?
  5. Figure 18: A curve for kst is missing (because it is out of range), but it is listed in the legend.
  6. Line 544: "At the beginning of loading..." It should be unloading.
  7. Line 546: "..final phase of loading..." Again, unloading.
  8. General: You studied eccentricity in a perpendicular direction only. What about the influence of eccentricity along the profiles?
  9. General: In the paper, experimental results have been presented for a specific geometry of the stack. The mathematical model is fully based on these results. Is there any wider applicability of your mathematical model? Namely, it makes no sense to model the phenomenon for which you already performed experimental tests, because you already have the results. Modelling to predict the unknown makes sense. Therefore, what is the wider impact of your research? I suggest to consider this question and include a corresponding discussion in the paper.

I suggest to improve the text as suggested and that all open items are clarified before resubmitting the manuscript.

Author Response

Authors are grateful to the Reviewer for an in-depth review and numerous remarks to the manuscript Smolana M., Gromysz K.: “Effect of Eccentricity of Applied Force and Geometrical Imperfections on Stiffness of Stack of Cuboidal Steel Elements”. Consideration of Reviewer comments will improve quality of the paper.

Below there are informations about the changes made to the manuscript in connection with the review. We also provide answers to the Reviewer's questions included in the review.

  1. (Reviewer) Line 134: The system was installed in the horizontal position. This neglects the influence of the dead weight of rolled profiles. Namely, in a vertical position, such a stack is axially loaded with dead weight as a body force. Please, comment on the neglect of this effect, especially at low values of the loading force.

Ref. to 1. (Authors) Four lines of text (lines 174-177) were added to the manuscript

In real situation, at the construction side, the stack of elements is installed in vertical position. The dead weight of 1 m high stack is approximately 3 kN. Neglecting during the test the values force Q within the range from 0 to 1.5 kN was in fact taking under consideration the dead weight of the stack.

  1. (Reviewer) Figure 4: How did you measure displacement and how did you exclude the influence of frame deformations? Please, add the comment about the importance of the frame rigidity.

Ref. to 2. (Authors) Three lines of text were added to manuscript (lines 147-149)

The measuring base used to determine changes in length ΔlA and ΔlB was attached to a stack of rectangular elements and had no contact with frame. So the frame rigidity did not affect the measured values of ΔlA and ΔlB.

  1. (Reviewer) Line 272: There is a strange definition for A ("area... acting at transferring"?). In eq. 6 it represents the area of cross-section, but it is difficult to define it at that point. Afterward, it is correctly used, but try to define it better.

Ref. to 3. (Authors) In line 279 of the manuscript the new definition for area A has been assumed: “cross-sectional area of diaphragm part of cuboidal element”

  1. (Reviewer) Table 3 and Figure 16a: Why is the slope of ea=40mm so low?

Ref. to 4. (Authors) For the eccentric case ea = 40 mm, the load path and unload path are close to each other. The hysteresis loop area is small. It means that internal friction of low value is occurring inside the cuboidal element.

  1. (Reviewer) Figure 18: A curve for kst is missing (because it is out of range), but it is listed in the legend.

Ref. to 5. (Authors) The stiffness kst is load-independent and has a constant value. Therefore in Figure 18 a straight line with a constant value of 6211 MN/m corresponds to the stiffness kst.

  1. (Reviewer) Line 544: "At the beginning of loading..." It should be unloading.

Ref. to 6. (Authors) The error has been corrected (from loading to unloading – line 555)

  1. (Reviewer) Line 546: "..final phase of loading..." Again, unloading.

Ref. to 7. (Authors) The error has been corrected (from loading to unloading – line 556)

  1. (Reviewer) General: You studied eccentricity in a perpendicular direction only. What about the influence of eccentricity along the profiles?

Ref. to 8. (Authors) It was decided to study the effect of eccentric in a perpendicular direction due to the fact that the impact of imperfections associated with inaccurate contact between cuboidal elements is the most visible in this direction. In addition, the stack has the smallest stiffness in transverse direction. It is therefore a direction with potentially the greatest sensitivity to the eccentric positioning of the jack. However, the study of the influence of the eccentric in the second direction is the subject of ongoing research.

  1. (Reviewer) General: In the paper, experimental results have been presented for a specific geometry of the stack. The mathematical model is fully based on these results. Is there any wider applicability of your mathematical model? Namely, it makes no sense to model the phenomenon for which you already performed experimental tests, because you already have the results. Modelling to predict the unknown makes sense. Therefore, what is the wider impact of your research? I suggest to consider this question and include a corresponding discussion in the paper.

Ref. to 9. (Authors) We believe that a model with a similar structure can also be adapted to other structures characterized by the occurrence of internal friction. The occurrence of "clearance" in structures is associated with additional deformations and related internal friction forces. The presented approach can also be used in simplified numerical calculations.

Currently, experimental studies are conducted on the impact of stack length on its stiffness.

It should also be noted that the tests already carried out have allowed the development of an improved support solution. Laboratory and in situ tests of this new solution are currently being carried out.

Reviewer 2 Report

The submitted draft titled ``Effect of Eccentricity of Applied Force and Geometrical Imperfections on Stiffness of Stack of Cuboidal Steel Elements'' is focused on a topic of interest in some specific applications.

The attention is to the role of imperfection and eccentricity in mounting hydraulic jack on stacks of cuboidal elements used in rectification processes of vertically deflected buildings.

Introduction and references therein are enough to give general ideas about the topic. Mathematical developments appear adequate to describe and analyze the problem.

Graphics are clear and readable.

The paper has an overall good quality. 'Research programme' (Sec.2) and 'Test procedure and results' (Sec.3) are maybe too long, although sufficiently clear.
'Model of the stack' (Sec.3) is the most interesting (in this reviewer's opinion). Is clear enough.
However, apparently it is not explained the reason to assume k_{con} as the quadratic function of Eq.(8).

The indefinite integral in Eq.(9) gives actually u_{con}(Q) plus an integration constant that seems computed at Q=0, where what the Authors call ``the boundary condition'' is met. There, the primitive function is set to the value '0'. If it is right, Eq.(10) should be written as:
u_{con}(0)=0,
that is u_{con}(Q)=0 is wrong.

Equation (9) could be written also as a definite integral, with lower limit 0 and upper limit Q (in such a case, Q inside the integral should be intended or replaced by a dummy variable). However, additional condition should be given (as those that are implicitly given on \Delta, in Eq.(12)).

In Eq.(11), the argument of logarithm is bracketed using straight brackets as ln|...|. Is is slightly confusing. Indeed it seems the argument of ln is an absolute value. While it is in general meaningful ln|...|=ln(|...|), here the argument of ln is not an absolute value. Hence, instead writing ln|...| it would be better ln(...).

Furthermore, Eq.(11) coul be written simpler as (here I use LaTeX symbols to write the formula):

u_{con}(Q)=(1/sqrt(Delta))ln((2 gamma_{con}+Q(beta_{con}+sqrt(Delta))/(2 gamma_{con}+Q(beta_{con}-sqrt(Delta)))

Unless I missed the point, I couldn't find estimated values of alpha_{con}, beta_{con}, gamma_{con}.

The remaining part of the paper appears clear enough.

Author Response

Authors are grateful to the Reviewer for an in-depth review and numerous remarks to the manuscript Smolana M., Gromysz K.: “Effect of Eccentricity of Applied Force and Geometrical Imperfections on Stiffness of Stack of Cuboidal Steel Elements”. Consideration of Reviewer comments will improve quality of the paper.

Below there are informations about the changes made to the manuscript in connection with the review. We also provide answers to the Reviewer's questions included in the review.

  1. (Reviewer) indefinite integral in Eq.(9) gives actually u_{con}(Q) plus an integration constant that seems computed at Q=0, where what the Authors call ``the boundary condition'' is met. There, the primitive function is set to the value '0'. If it is right, Eq.(10) should be written as:
    u_{con}(0)=0,
    that is u_{con}(Q)=0 is wrong.

Ref. to 1. (Authors) Formula (10) has been changed to the one proposed by the Reviewer.

  1. (Reviewer) Equation (9) could be written also as a definite integral, with lower limit 0 and upper limit Q (in such a case, Q inside the integral should be intended or replaced by a dummy variable). However, additional condition should be given (as those that are implicitly given on \Delta, in Eq.(12)).

Ref. to 2. (Authors) Formula (10) has been changed to the one proposed by the Reviewer. Quantity Q inside the integral has been replaced by a dummy variable Q*.

  1. (Reviewer) In Eq.(11), the argument of logarithm is bracketed using straight brackets as ln|...|. Is slightly confusing. Indeed it seems the argument of ln is an absolute value. While it is in general meaningful ln|...|=ln(|...|), here the argument of ln is not an absolute value. Hence, instead writing ln|...| it would be better ln(...).

Ref. to 3. (Authors) Formula (11) has been changed to the one proposed by the Reviewer.

  1. (Reviewer) Furthermore, Eq.(11) coul be written simpler as (here I use LaTeX symbols to write the formula): u_{con}(Q)=(1/sqrt(Delta))ln((2 gamma_{con}+Q(beta_{con}+sqrt(Delta))/(2 gamma_{con}+Q(beta_{con}-sqrt(Delta)))

Ref. to 4. (Authors) The form of formula (11) has been left as in Note 4

  1. (Reviewer) Unless I missed the point, I couldn't find estimated values of alpha_{con}, beta_{con}, gamma_{con}.

Ref. to 5. (Authors)  The values of the coefficients αcon, βcon, and γcon are in Table 2. To increase clarity, the heading in Table 2 has been changed.

Reviewer 3 Report

The authors present a model for determining stiffness of a stack of cuboidal steel elements used for building rectification. Experimental test with cyclic loading and varying eccentricity of the load is conducted on a laboratory stack with 27 elements. Three stiffness terms are modeled, including one linear elastic stiffness and two stiffnesses related with geometrical imperfections in the cuboidal elements. The proposed model is interesting and the predicted hysteresis loops obtained from the suggested model shows a very good match with experiment. Nevertheless, the authors need to make clear some points as following:

  1. The stack is modeled as in Figure 3(b). How do you confirm the rigid connection between the stack support and the base?
  2. The free-ends rod of double the length model is used to replace the real one-end fixed rod model. Then, I think additional explanations on the equivalent between the two models are needed.
  3. Do you consider the effect of deformation of the supporting steel frame?
  4. It is not clear how is equation (1) built?
  5. If jack (5) in Figure 4 is a passive jack, so why is it needed. Perhaps, it can be replaced by other simpler mechanical supporting structure.
  6. How many tests are repeated for one experiment?
  7. Page 10: “Fig. 22b” should be “Fig. 2b”.
  8. The imperfection concerned deviation in dimensions and production inaccuracies of cuboidal elements uint is assumed in Fig. 10c. In practice, there are many possible dimensional deviations caused by production inaccuracies. So why do you consider this imperfection. Does it the most significant one compared to the others?
  9. Page 13 lines 300-302: It is confusing that you mention non-elastic element Nint,fr, elastic force Nint, fr, and also friction force Nint,fr.
  10. Appropriate references (citations) for equations (8) and (13) are needed.
  11. There is a little confusing in the structure of the model in section 4.1. Displacement of the compression phase II is represented by equation (2) as u = ust + ucont + uint. The invariant stiffness kst is calculated using model presented in Figure 4, which assumes that all the vertical parts of cuboidal element are under loading. Then, the contacting imperfection ucon, which is represented by a gap as in Fig. 10b, would be diminished. But why does ucon still appear simultaneously with ust in equation (2)?
  12. Section 5.2: Phase II is modeled with two springs kcon and kst. But in this phase, the friction obviously occurs. Why don’t you consider friction in (33) of this phase model?
  13. Figures 16: Position of the schematic diagram of the model can be adjusted to avoid overlapping the main plotted lines.
  14. 18: Although the authors have determined the stiffness kint for various eccentricities, it is not clear how the real geometric imperfections (Fig. 10b and c) affect the stiffness. Additional discussions are expected.
  15. The limitation of proposed model needs to be discussed further. The present model is based on a laboratory size stack. Can the suggested stiffness formulation be adopted for larger stack, which is used in actual building rectification?

Author Response

Authors are grateful to the Reviewer for an in-depth review and numerous remarks to the manuscript Smolana M., Gromysz K.: “Effect of Eccentricity of Applied Force and Geometrical Imperfections on Stiffness of Stack of Cuboidal Steel Elements”. Consideration of Reviewer comments will improve quality of the paper.

Below there are informations about the changes made to the manuscript in connection with the review. We also provide answers to the Reviewer's questions included in the review.

  1. (Reviewer) The stack is modeled as in Figure 3(b). How do you confirm the rigid connection between the stack support and the base?

Ref. to 1. (Authors) A rigid connection between the stack support and the base occurs if the force Q acts on the eccentric less than h/6 (h - stack cross-section height) relative to the base of the stack and there are no imperfections in the base. In other cases there is no rigid connection. Therefore, in Figure 3 (b) the rigid connection symbol has been removed.

  1. (Reviewer) The free-ends rod of double the length model is used to replace the real one-end fixed rod model. Then, I think additional explanations on the equivalent between the two models are needed.

Ref. to 2. (Authors) Adopting a substitute system for research, it was strived to ensure that the deformations of the substitute system and the real system were equal. In particular, maximum displacements corresponding to 2nd order deformations should be equal.

To demonstrate the correctness of the adopted procedure one specific case of the static schema will be examined. It is assumed one-end fixed rod model of the stack of the length l and of constant stiffness EI of the cross-section. A relationship is obtained for the maximum displacements of the second order of the real system

This equation can be approximated by a relationship

where Qcrit is the critical load value.

Qcrit in the case of the real model (cantilever system with length l) is:

It means

In the case of free-ends rod of double the length model

It means

The above means that the second order displacements of the real and substitute systems are the same. Demonstration of the above for other operating conditions would require the use of numerical methods.

It is possible to proof that in case of other static schemes of stack of elements (schemes discussed in Gromysz, K. Analysis of the effect of load application eccentricity on the stiffness of supports consisting of stack of elements. MATEC Web Conf. 2019, 262, 10005, doi:10.1051/matecconf/201926210005) the second order displacements of the real and substitute systems are the same.

  1. (Reviewer) Do you consider the effect of deformation of the supporting steel frame?

Ref. to 3. (Authors) Three lines of text were added to manuscript (lines 147-149)

The measuring base used to determine changes in length ΔlA and ΔlB was attached to a stack of rectangular elements and had no contact with frame. So the frame rigidity did not affect the measured values of ΔlA and ΔlB.

  1. (Reviewer) It is not clear how is equation (1) built?

Ref. to 4. (Authors) The change in the length of the stack in the load action axis was determined by linear interpolation between the values measured at the edges of the stack.

The above information has been added to the manuscript (lines 145-146)

  1. (Reviewer) If jack (5) in Figure 4 is a passive jack, so why is it needed. Perhaps, it can be replaced by other simpler mechanical supporting structure.

Ref. to 5. (Authors) The use of a passive jack ensures the symmetry of the tested system. This is a necessary condition to reproduce the working conditions of the real support using equivalent system.

  1. (Reviewer) How many tests are repeated for one experiment?

Ref. to 6. (Authors) Four full load-unload cycles were performed for one experiment (research program was presented in Fig. 6). Measurement results (four loops) corresponding to maximum and minimum values Qmax for an eccentricity value ea are presented in Fig. 8. No significant differences were observed between the hysteresis loops in the given experiment. For this reason, when determining model parameters one loop was considered later in the manuscript

  1. (Reviewer) Page 10: “Fig. 22b” should be “Fig. 2b”.

Ref. to 7. (Authors) The error has been corrected by changing “Fig. 22b” to “Fig. 2b” (line 223).

  1. (Reviewer) The imperfection concerned deviation in dimensions and production inaccuracies of cuboidal elements uint is assumed in Fig. 10c. In practice, there are many possible dimensional deviations caused by production inaccuracies. So why do you consider this imperfection. Does it the most significant one compared to the others?

Ref. to 8. (Authors) The described imperfection is large enough to be visible to the naked eye. Displacement uint, however, can be identified with a whole group of imperfections, as a result of which there are mutual displacements of profiles inside a single rectangular element when loading the stack. The drawing was mainly intended to capture the nature of these imperfections.

Above information was added in lines 304-306

  1. (Reviewer) Page 13 lines 300-302: It is confusing that you mention non-elastic element Nint,fr, elastic force Nint,fr, and also friction force Nint,fr.

Ref. to 9. (Authors) Lines 310-312 (in the previous version of the manuscript, lines 300-302) have been reworded. Currently they read as follows:

“Therefore, the non-elastic element connected in parallel with the spring kint is inserted into the model. The external load Q was balanced by two internal forces: the elastic force Nint,el present in the spring, and the friction force Nint,fr in the non-elastic element.”

  1. (Reviewer) Appropriate references (citations) for equations (8) and (13) are needed.

Ref. to 10. (Authors) Forms of equations (8) and (13) were adopted by the authors based on the analysis of measured results of phases I and II. This was additionally indicated in the text in line 291 and in line 315.

  1. (Reviewer) There is a little confusing in the structure of the model in section 4.1. Displacement of the compression phase II is represented by equation (2) as u = ust + ucont + uint. The invariant stiffness kst is calculated using model presented in Figure 4, which assumes that all the vertical parts of cuboidal element are under loading. Then, the contacting imperfection ucon, which is represented by a gap as in Fig. 10b, would be diminished. But why does ucon still appear simultaneously with ust in equation (2)?

Ref. to 11. (Authors) Ust represents the displacement that would occur in an idealized stack that has no imperfections. Real stack displacements under increasing load are increased by closing the free spaces between the cuboidal elements. The closing of free spaces can therefore be presented as additional displacements denoted as ucon. Assuming this assumption, both displacements ust and ucon occur at each load change.

  1. (Reviewer) Section 5.2: Phase II is modeled with two springs kcon and kst. But in this phase, the friction obviously occurs. Why don’t you consider friction in (33) of this phase model?

Ref. to 12. (Authors) In Phase II friction occurs. However, it has the nature of static friction in this phase. Static friction, prevents uint displacements. The moment when static friction is overcome, Phase III begins. In phase III, the friction already takes on a kinematic character and uint displacements occur again.

  1. (Reviewer) Figures 16: Position of the schematic diagram of the model can be adjusted to avoid overlapping the main plotted lines.

Ref. to 13. (Authors) Changes have been made to the graphs from Figure 16

  1. (Reviewer) Although the authors have determined the stiffness kint for various eccentricities, it is not clear how the real geometric imperfections (Fig. 10(b) and (c)) affect the stiffness. Additional discussions are expected.

Ref. to 14. (Authors) New lines 516-521 and new figure (Fig 21) were added to the manuscript to clear how the real geometric imperfections affect the stiffness.

The effect of imperfection of inaccurate contact between the stack elements of (kint) on the equivalent stack stiffness kst,int was described by the ratio kst,int/kst (definition of kst,int see equation (36) in the manuscript). It’s value is changing almost linearly in a function of the load Q. For Q = 0, its value was close to zero, and for Qmax = 1000 kN the value was equal to 0.16.

  1. (Reviewer) The limitation of proposed model needs to be discussed further. The present model is based on a laboratory size stack. Can the suggested stiffness formulation be adopted for larger stack, which is used in actual building rectification?

Ref. to 15. (Authors) Currently, experimental studies are conducted on the impact of stack length on its stiffness.

It should also be noted that the tests already carried out have allowed the development of an improved support solution. Laboratory and in situ tests of this new solution are currently being carried out.

Round 2

Reviewer 1 Report

The authors have correctly and consistently considered all the comments, so I recommend the manuscript for publication.